# Investigating Continual Pretraining in Large Language Models: Insights and Implications

**Çağatay Yıldız**  *cagatay.yildiz@uni-tuebingen.de*
*University of Tübingen*

**Nishaanth Kanna Ravichandran**  *nishaanthkanna@gmail.com*
*Cohere for AI Community*

**Nitin Sharma**  *nitinsharma3150@gmail.com*
*University of Tübingen*

**Matthias Bethge**  *matthias.bethge@uni-tuebingen.de*
*Tübingen AI Center, University of Tübingen*

**Beyza Ermis**  *beyza@cohere.com*
*Cohere for AI*

**Reviewed on OpenReview:** `https://openreview.net/forum?id=aKjJoEVKgO`

## Abstract

Continual learning (CL) in large language models (LLMs) is an evolving domain that focuses on developing efficient and sustainable training strategies to adapt models to emerging knowledge and achieve robustness in dynamic environments. Our primary emphasis is on *continual domain-adaptive pretraining*, a process designed to equip LLMs with the ability to integrate new information from various domains while retaining previously learned knowledge. Since existing works concentrate mostly on continual fine-tuning for a limited selection of downstream tasks or training domains, we introduce a new benchmark designed to measure the adaptability of LLMs to changing pretraining data landscapes. We further examine the impact of model size on learning efficacy and forgetting, as well as how the progression and similarity of emerging domains affect the knowledge transfer within these models.

Our findings uncover several key insights: (i) continual pretraining consistently improves $< 1.5B$ models studied in this work and is also superior to domain adaptation, (ii) larger models always achieve better perplexity than smaller ones when continually pretrained on the same corpus, (iii) smaller models are particularly sensitive to continual pretraining, showing the most significant rates of both learning and forgetting, (iv) continual pretraining boosts downstream task performance of GPT-2 family, (v) continual pretraining enables LLMs to specialize better when the sequence of domains shows semantic similarity while randomizing training domains leads to better transfer and final performance otherwise. We posit that our research establishes a new benchmark for CL in LLMs, providing a more realistic evaluation of knowledge retention and transfer across diverse domains.

## 1 Introduction

Recent advancements in the field of Natural Language Processing (NLP) have been significantly shaped by the development of large language models (LLMs) (Devlin et al., 2018; Radford et al., 2019; Brown et al., 2020). These models, trained on vast corpora from diverse domains, have emerged as versatile tools for numerous NLP tasks. However, the increasing scale and complexity of LLMs have raised concerns about the financial and ecological costs associated with training them from scratch (Luccioni et al., 2022). This has necessitated more efficient approaches than retraining these models with each new data stream. Continual Learning (CL) emerges as a crucial strategy (Sun

Table 1: The details of the L1 domains used in our experiments. Note that Art and Philosophy did not have any subdomains in M2D2 dataset.

| L1 DOMAIN (ABBRV) | SIZE | #L2 | #TOKENS | EXAMPLES OF L2 DOMAINS |
|---|---|---|---|---|
| Culture and The Arts (Culture) | 1.8 GB | 7 | 265M | Mass Media, Sports and Recreation |
| History and Events (History) | 1.2 GB | 3 | 208M | Region, Period |
| Technology and Applied Sciences (Tech) | 1.7 GB | 4 | 268M | Agriculture, Computing |
| Health and Fitness (Health) | 739 MB | 6 | 99M | Exercise, Nutrition |
| Religion and belief systems (Religion) | 341 MB | 3 | 48 M | Belief Systems, Major beliefs |
| General reference (GeneralRef) | 196 MB | 2 | 39M | Reference works |
| Philosophy and thinking (PhilThink) | 721 MB | 2 | 124M | Philosophy, Thinking |
| Art | 578 MB | 1 | 98 M | – |
| Philosophy | 919 MB | 1 | 156M | – |
| Quantitative Biology (Bio) | 1.9 GB | 11 | 336M | Biomolecules, Cell Behavior |
| Physics | 4.1 GB | 22 | 737M | General Physics, Biological Physics |
| Condensed Matter (CondMat) | 3.5 GB | 9 | 570M | Materials Science, Quantum Gases |
| Nonlinear Sciences (Nlin) | 730 MB | 5 | 134M | Self-Organizing Systems |
| Mathematics (Math) | 4.5 GB | 30 | 1.4B | Topology, Number Theory |
| Statistics (Stat) | 2.4 GB | 6 | 450M | Applications, Methodology |
| Economics (Econ) | 67 MB | 3 | 11M | Econometrics, Theory |
| Computer Science (CS) | 4.5 GB | 39 | 1.1B | Machine Learning, Graphics |
| Astrophysics (Astro) | 3.1 GB | 5 | 562M | Earth/Planetary, Cosmology |
| Total | 32.4 GB | 159 | 6.6B | – |

et al., 2019; Biesialska et al., 2020) to reduce both financial and environmental costs while keeping models up-to-date. A critical aspect of this adaptation is ensuring that knowledge transfer occurs seamlessly across domains without catastrophic forgetting (French, 1999) and operates effectively without explicit domain identification for each task.

Continual learning works in LLMs can broadly be divided into two sub-categories: *continual fine-tuning* and *continual domain-adaptive pretraining*. The former incrementally fine-tunes an LLM on a sequence of downstream tasks (Wu et al., 2021; Ramasesh et al., 2021; Scialom et al., 2022; Mehta et al., 2023; Gururangan et al., 2021; Khan et al., 2022; Zhang et al., 2022; Razdaibiedina et al., 2023; Luo et al., 2023b). In this work, we focus on the latter - adapting LLMs to new domains through incremental updates, thereby avoiding the need for exhaustive retraining whenever new data becomes available (Xu et al., 2019; Gururangan et al., 2020; Ke et al., 2023b). Despite growing interest in this area, prior studies have primarily examined small-scale models or a limited number of training domains (Qin et al., 2022; Gupta et al., 2023; Luo et al., 2023a; Ke et al., 2023b; Gogoulou et al., 2024; Cossu et al., 2024) (see Wu et al. (2024) for a recent survey and Section 5 for related works). The most relevant work, Gururangan et al. (2020), evaluated the transfer capabilities of a RoBERTa model continually pretrained across four large domains. However, a comprehensive assessment of forgetting and knowledge transfer in LLMs across diverse architectures and scales remains lacking. Furthermore, conventional CL benchmarks such as split CIFAR-100 (Krizhevsky et al., 2009) and Tiny ImageNet (Le & Yang, 2015) operate at a much smaller data and model scale and thereby fail to capture the unique challenges of continual pretraining in LLMs.

To bridge this gap, we conduct an extensive benchmark by pretraining LLMs across a wide range of domains and evaluating their performance throughout the learning process. Unlike prior works that focus on a narrow set of domains, our study leverages the Massively Multi-Domain Dataset (M2D2) (Reid et al., 2022), which spans 236 hierarchically organized domains from Wikipedia and Semantic Scholar, enabling a detailed investigation of forgetting and knowledge transfer in a large-scale setting. Our study systematically explores the core dynamics of continual pretraining in LLMs by *(i)* comparing continual pretraining vs. standalone adaptation across various architectures and model scales, *(ii)* analyzing forward and backward knowledge transfer and forgetting at different stages of continual learning, and *(iii)* investigating how perplexity is influenced by the order of training domains, batch size, and data imbalance. By quantifying key CL metrics, we reveal fundamental differences between LLMs and traditional small-scale CL setups, emphasizing that insights from prior CL research may not generalize to large-scale continual pretraining. Rather than proposing solutions to mitigate forgetting, our goal is to characterize the unique challenges of CL in LLMs and provide a realistic benchmark for future studies. Our findings uncover several key insights:

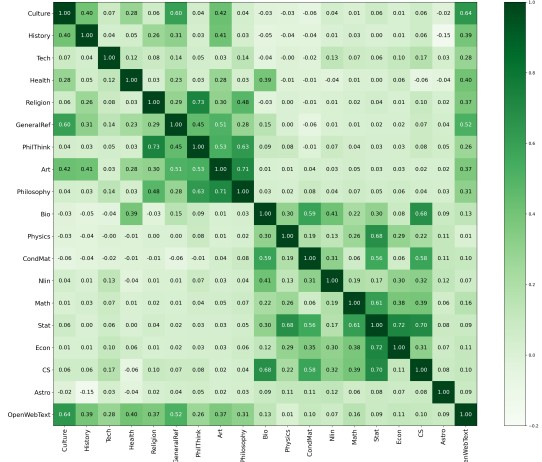 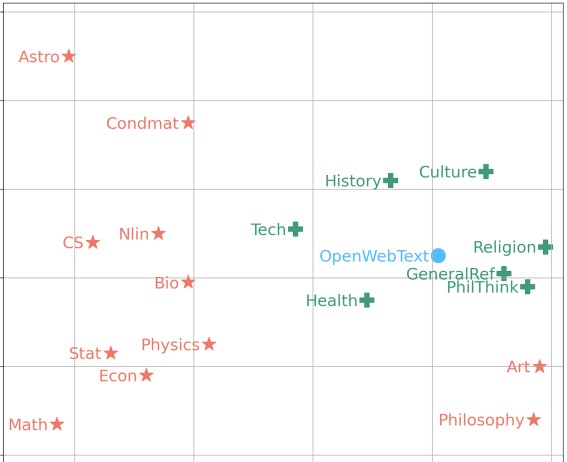

Figure 1: Cosine similarity between our L1 training domains. We also include OpenWebText (Gokaslan & Cohen, 2019), an open-source replication of the GPT2 pretraining data set. The two big square blocks along the diagonal correspond to Wiki and S2ORC portions.

Figure 2: Average L1-domain embeddings visualized using t-SNE. Wiki domains and natural sciences form two clear clusters. Note that Art and Philosophy are from S2ORC portion, but they are closer to Wiki due to they are social sciences and the rest of S2ORC is natural sciences.

    i) Continual pretraining improves GPT2 models of all sizes while Llama2-7B does not benefit from it. This is because the majority of the training domains are too small for Llama2-7B to be adapted to.

    ii) Continual pretraining outperforms domain-adaptive pretraining across all model families and sizes. Importantly, pretraining on related domains enhances knowledge transfer, leading to better specialization when developing expert models for a given domain.

    iii) Final model performance correlates positively with model size. Surprisingly, smaller models consistently show the highest degree of learning and forgetting.

    iv) Randomizing the order of training domains enables positive transfer and reduced forgetting.

    v) Continual pretraining curriculum determines the downstream task performance.

## 2  Methodology

In this section, we describe our training process, provide an overview of the tasks (domains) used for continual pretraining and assessment of models, and explain the evaluation pipeline.

**Training**     We initiate our process with a pretrained LLM that has been already trained on a comprehensive corpus, which generally represents a broad or general domain, such as a book corpus or web content. We then perform continual pretraining on a series of unlabeled domain corpora. Our goal is to continuously pretrain an LLM on these sequential domain corpora by using the original training objectives, e.g., the next token prediction likelihood for autoregressive LLMs. Note that once a domain corpus is used for training, it is no longer available.

**Tasks**     Our experiments are conducted on the M2D2 dataset (Reid et al., 2022), which is an extensive and finely categorized corpus specifically designed for exploring domain adaptation in language models. It comprises 8.5 billion tokens and covers 236 domains, sourced from Wikipedia and the Semantic Scholar (S2ORC) database Lo et al. (2019). This dataset is unique in its combination of fine domain granularity and a human-curated domain hierarchy, set within a multi-domain context.

The corpus is divided into two levels: L1-domains and L2-domains. In the context of the S2ORC corpus, L1-domains refer to broad fields of academic research, such as Computer Science and Physics, while L2-domains correspond to

specific arXiv categories within these fields, like "Artificial Intelligence" under Computer Science. For Wikipedia, L1-domains represent major categories, and L2-domains encompass category pages within each L1 domain. To maintain balance and computational efficiency in our experiments, we excluded domains exceeding 5GB of data, such as Medicine. Ultimately, we utilized 159 domains in our study (see Table 1 for details).

To show the cross-domain similarity, we first computed the task embedding by using Sentence-BERT (Reimers & Gurevych, 2019) with 10K samples from each domain and 50K samples from OpenWebText (Gokaslan & Cohen, 2019), an open-source reproduction of GPT2 training dataset (Radford et al., 2019). Then we computed cosine similarities between each task pair (Figure 1). For the *similar-order* experiments detailed in the next section, we order the training domains based on their similarity, starting with the Culture domain, which is the most similar to OpenWebText, and then proceeding to the next domain by sampling from the remaining ones based on similarity. Also see Figure 2 for the average L1 embeddings visualized using t-SNE.

**Evaluation**  Each domain in the M2D2 dataset is split into train, validation, and test sets with no data leakage, as outlined in Reid et al. (2022). Each validation and test set includes over 1 million tokens, allowing accurate evaluations within specific domains. We measure the effectiveness of all methods by testing perplexity on L2 domain test sets. For continual domain-adaptive pretraining experiments, after completing training on a domain for one epoch, we checkpoint the model, and compute the test perplexity for current and previous domains.

## 3  Experimental Setup

**Models and training**  We benchmark continual learning of existing pretrained LLMs with different architectures and sizes. In particular, we consider *(i)* decoder-only models (GPT2-S, GPT2-M, GPT2-L and GPT2-XL, Llama2-7B) as well as *(ii)* encoder-decoder models (RoBERTa-base and RoBERTa-large (Liu et al., 2020)). We trained the models with Adam optimizer (Kingma & Ba, 2015) with a batch size of 16 sequences on NVIDIA A100 GPUs. We used DeepSpeed (Rasley et al., 2020) with auto configuration, which assigns a dropout rate of 0.2 and automatic learning-rate selection.

**Task ordering**  In order to investigate how the order of training domains impacts our domain-incremental continual learning setup, we ordered the tasks in our experiments in two different ways: *(i) similar-order* where semantically related domains follow one another as described in the previous section, and *(ii) random-order*, where the domains are shuffled.

**Metrics for assessing continual learning efficacy**  To evaluate the effectiveness of continual learning, we begin by setting two baselines for comparison: *zero-shot perplexity (ZS)* which measures the innate ability of the original, unmodified models to predict outcomes without any domain-specific tuning, and *domain adaptive pretraining perplexity (DAPT)*, where we first pretrain original models on a particular domain and then evaluate it on the test portion of the training domain. In the same vein as Gururangan et al. (2020), our continued pretraining on a corpus of unlabeled domain-specific text is arguably expected to lead to better expertise on that particular domain. *ZS* acts as a fundamental baseline, ensuring that our models have a basic level of competence and *DAPT* sets a targeted performance standard for our continual learning approach to surpass. Achieving a better perplexity than the *DAPT* baseline is the primary objective for continual pretraining, signifying that longer training horizons are more favorable than stand-alone pretraining that adapts a model to a single domain.

To assess continual learning performance, we compute *continual pretraining perplexity (CPT)* where we evaluate the performance of a model $\mathcal{M}_i$ on the most recent training domain $\mathcal{D}_i$. This measure helps us understand how well the model adapts to new information over time. Moreover, we compute the *last checkpoint (LC)* $\mathcal{M}_N$ against all the training domains to examine the final model's ability to retain and transfer knowledge across a broad range of subjects. We calculate *forgetting (FG)* on a previous domain by taking the difference between the perplexity of the current checkpoint on that domain and the best perplexity achieved on that domain so far. Finally, we evaluate checkpoints on previously seen/unseen domains to measure backward/forward transfer.

**Metric expressed explicitly**  Let $\mathcal{M}_0$ denote the pretrained LLM that has been already trained on a comprehensive corpus $\mathcal{D}_0$. Our continual pretraining corpora $\mathcal{S}_N = \{\mathcal{D}_1, \cdots, \mathcal{D}_N\}$ consists of $N$ domains. At each stage $i$, the

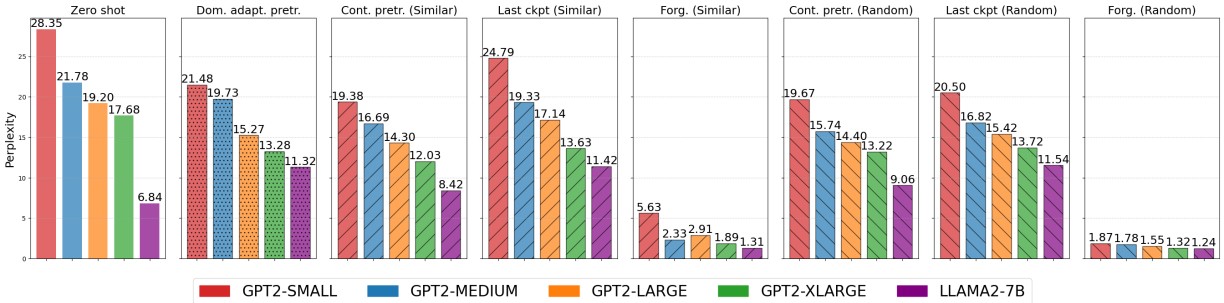

Figure 3: Above panels show test perplexities ($\downarrow$) with different model sizes and training orders. For reference, we include the zero-shot and domain adaptation perplexities. Please see Figure 16 for results obtained on Wiki and S2ORC domains.

LLM $\mathcal{M}_{i-1}$ is trained on a new corpus $\mathcal{D}_i$, resulting in an updated model $\mathcal{M}_i$. Hence, our continual learning setup generates $\mathcal{M}_{1:N}$ checkpoints. Let $z_n$ and $f_n$ denote the zero-shot (ZS) and domain adaptive pretraining perplexities (DAPT) computed on $n$'th domain. Note that $z_n$ is obtained by evaluating $\mathcal{M}_0$ on $\mathcal{D}_n^{\text{test}}$ whereas DAPT involves first training $\mathcal{M}_0$ on $\mathcal{D}_n^{\text{tr}}$ and then testing on $\mathcal{D}_n^{\text{test}}$. Further, let $p_n^c$ denote the perplexity of $c$'th checkpoint $\mathcal{M}_c$ on $\mathcal{D}_n$ (notice that $c > n$ and $c < n$ correspond to backward and forward transfer). Then the main metrics of our interest are computed as follows: $\text{ZS} = \texttt{median}(z_{1:N})$, $\text{DAPT} = \texttt{median}(f_{1:N})$, $\text{CPT} = \texttt{median}(p_1^1, \dots, p_N^N)$, $\text{LC} = \texttt{median}(p_{1:N}^N)$, $\text{FG} = \texttt{median}(g_2, \dots, g_N), g_c \equiv p_n^c - \min(p_n^{1:c-1})$. Note that checkpoints sometimes achieve arbitrarily high perplexity on a single domain while perplexities on other domains are within the usual range. Hence, we chose median for aggregation instead of mean to deal with such outliers. We also verify that median and mean aggregations differ by less than 1 percent except the outlier samples.

# 4 Findings and Analysis

In this section, we discuss our main findings. We first discuss the main results (Section 4.1), followed by how the model scale impacts continual learning (Section 4.2). Section 4.3 examines the implications of the order of training domains. The subsequent Section 4.4 discusses our positive forward transfer findings. Then we analyze fine-tuning performances on benchmark tasks in Section 4.5 and Section 4.6 connects these findings with a novel analysis based on prediction ranks. Finally, we list our remaining observations and ablation studies (Section 4.7).

## 4.1 What is the efficacy of continual learning?

**Continual pretraining consistently improves GPT2 family** We start by investigating whether continual learning is beneficial or not. Comparing the "last checkpoint" panels against "zero-shot", we observe that GPT2 models obtained at the end of continual learning always achieve better perplexity on the trained domains, implying significant knowledge accumulation. The overall improvement is overall more pronounced in random-order training. Further, the perplexities achieved by "continual pretraining" are always superior to the ones by "domain adaptive pretraining". Hence, in order to obtain a collection of expert models on a diverse range of domains, continual learning and checkpointing at domain shifts is a superior strategy to adapting a base model to each domain separately.

**Llama2-7B perplexity does not improve by CL since the domains are too small** When we turn to the Llama2-7B results, we most strikingly notice that additional training (including domain adaptation) always degrades the perplexity, which conflicts with the GPT2 findings presented above. Diving deeper into the model details, we notice that the largest GPT2 variant (GPT2-XL) and Llama2-7B have comparable model sizes and architectures whereas their training data substantially differ: GPT2-XL is trained on 9B tokens data curated from Web pages while Llama2-7B has a foundational, Internet-scale training dataset with 2T tokens. Hence, we conjecture that additional domain adaptive and/or continual pretraining on a relatively small corpus causes Llama2-7B to perform worse. Please note that we did not observe any training issues during learning, i.e., training perplexity always improved.

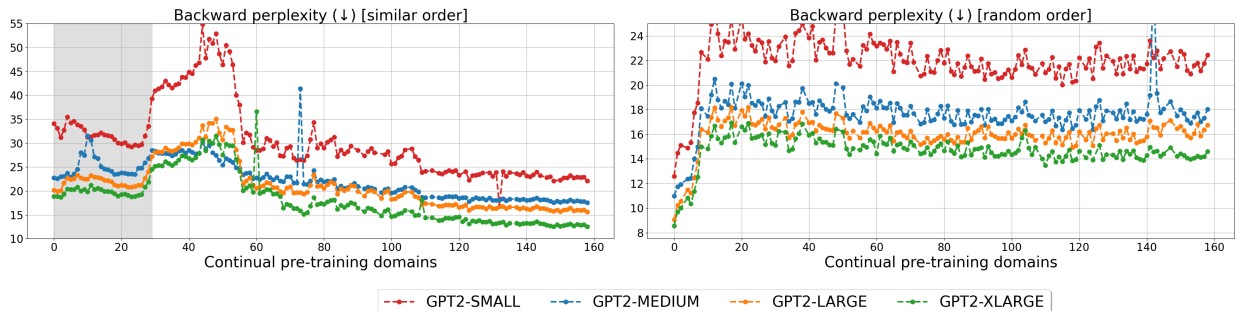

Figure 4: Median backward transfer perplexity during continual pretraining. The grey background highlights Wiki domains. Larger models always achieve better perplexity, which is also aligned with the initial (zero-shot) perplexity. Please see Section 4.2 for a detailed analysis.

To inspect our hypothesis, we investigate how domain size impacts the improvement of domain adaptive pretraining for Llama2-7B and GPT2-XL. Each point in Figure 18 corresponds to one domain adaptation experiment. The $x$-axis corresponds to domain sizes and the $y$-axis is the perplexity after domain adaptation (normalized by zero-shot). Positive $y$ values are the domains on which domain adaptive pretraining degrades perplexity. The left panel shows that Llama2-7B perplexity very rarely improves if the domain is smaller than 75 MB and it converges to zero after 100 MB. On the contrary, additional pretraining improves GPT2-XL perplexity on all but six domains. These findings imply that additional pretraining of Llama2-7B model is useful only for relatively larger domains.

## 4.2   How does model scale impact learning, forgetting, and final performance?

**Larger models always perform the best**   In agreement with the recent research on scaling laws (Kaplan et al., 2020; Bahri et al., 2021), Figure 3 shows that the model size perfectly correlates with the model performance. This holds true for all metrics (DAPT, CPT, LC) and both orders (with random order achieving better final performance). Further, for each checkpoint obtained after training on an L2 domain, we report the median backward perplexity (computed on all previous domains), and visualize it in Figure 4. The figure reveals that the backward perplexity at any stage of the training correlates with the model size.

**Smaller models benefit more from continual learning**   In relation to the previous analysis, we investigate how much CL improves overall model performance. For this, we subtract zero-shot perplexities from perplexities achieved during CL and visualize the results in Figure 5. We can see for random-order training that the improvement is inversely correlated with the model size, which we find intuitive since larger models, already achieving good perplexity, have less room for improvement. The curves corresponding to similar-order training (Figure 5, left) are rather cluttered; therefore, we inspect the *average* improvement over learning trajectories. The averages show the same performance improvement trend (GPT2-S > GPT2-M > GPT2-L) while GPT2-XL surprisingly comes at the second place.

**Forgetting inversely correlates with model size**   Finally, we quantify how much models forget due to continual learning. The "forgetting" panels in Figure 3 show that the smallest/largest model forgets the most/least while GPT2-M and GPT2-L are flipped when the training order changes. While following the trend, we believe Llama2-7B forgetting numbers should be taken with a grain of salt since perplexity does not improve overall. Nevertheless, overall, we conclude that increasing model size could be a way to alleviate forgetting although even the largest model forgets to some extent.

## 4.3   When does training in similar-order help?

As touched upon previously, Figures 3-5 demonstrate the positive impact of randomized training order on the model performance and transfer. While these findings consistently favor random order training, CPT panels in Figure 3 exhibit an interesting pattern: larger models seem to attain better CPT perplexity in similar order training than random order.

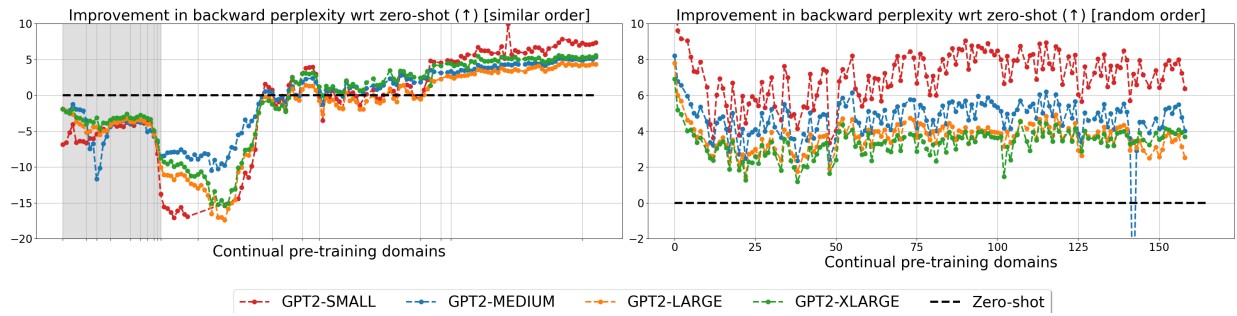

Figure 5: Median *improvement* in the backward transfer perplexity, normalized by zero-shot perplexity. The grey background highlights Wiki domains. Values smaller than zero indicate negative backward transfer. The backward transfer is at its lowest when the portions are switched in similar-order training (left), then it improves again. Positive backward transfer remains throughout learning in random-order.

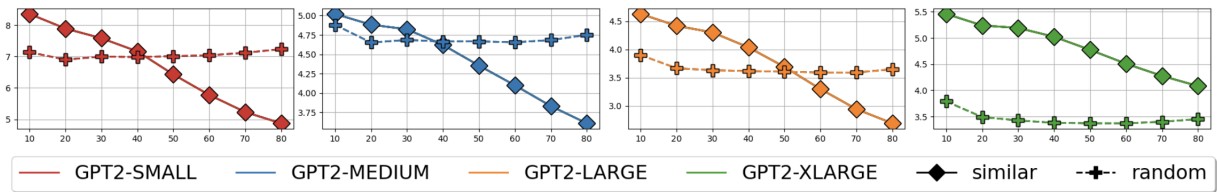

Figure 6: Median improvement in backward perplexity (normalized by zero-shot, $y$ axes, higher is better) as a function of the number of tasks between the checkpoint and the tested domain ($x$ axes). The diamonds and circles correspond to the similar and random order training and colors denote the models. The benefits of continual learning in similar-order steadily degrade with the transfer distance while it is stable for random order.

To further analyze this, instead of reporting an aggregated performance on all previous domains, we contrast the perplexity change with how old the evaluation domains are. In particular, $x$-axis of Figure 6 shows how many tasks have passed between a checkpoint and a domain it is tested on. For all model sizes, similar-order training impacts backward transfer to *recent past* (up to 40 domains) more positively than random-order. Naturally, the improvement in the case of similar-order training degrades over time since the recent training domains become significantly dissimilar to tested domains while random-order training does not worsen. Also, very interestingly, similar-order training in larger models seems to degrade less, indicating less forgetting, which agrees with the finding in the previous section that *larger models forget less*.

## 4.4 When do we observe positive forward transfer?

Our analysis so far has focused mainly on backward transfer and forgetting. In this section, we investigate whether models exhibit positive forward transfer by evaluating models on future domains.

**Random-order training consistently leads to positive forward transfer**    For our first analysis, we evaluate all GPT2-S checkpoints saved after L2 domains on all unseen domains. Figure 7 shows how forward transfer perplexity improves upon zero-shot performance. Noticeably, positive forward transfer is possible only to the S2ORC portion, i.e., no perplexity improvement when the model is evaluated on the Wikipedia portion. This discrepancy is expected as the S2ORC portion is about five times larger than the Wikipedia portion. Further, test perplexity on the S2ORC portion consistently improves with the number of pretraining tasks, i.e., longer training improves forward transfer. We conclude that the model accumulates knowledge that is on average beneficial to predict next tokens on unseen finer-grained domains.

**Positive forward transfer in similar training order is possible to only semantically related domains**    To examine how training on similar-order domains affects forward transfer, we evaluate model checkpoints on all subsequent L1

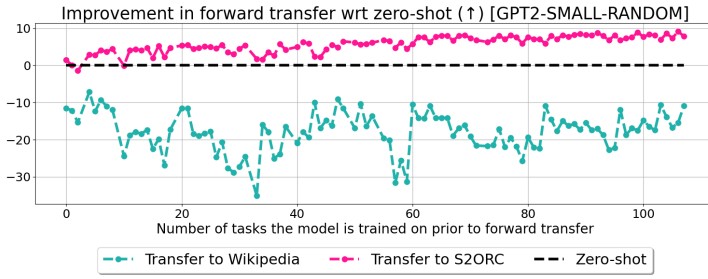
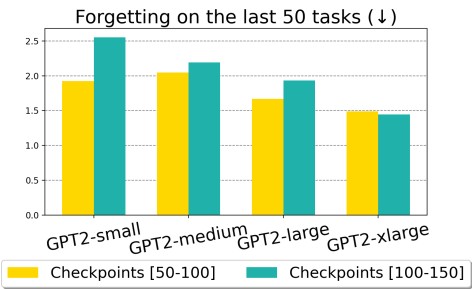

Figure 7: Forward transfer results with random training order. The $x$ axis shows the number of domains the model is trained on before forward transfer. Curves show the perplexity improvement over zero-shot. Clear positive/negative forward transfer to S2ORC/Wiki portions is observed.

Figure 8: We divide the checkpoints in random-order training into two groups based on their recency, showing that earlier checkpoints transfer better to the recent past.

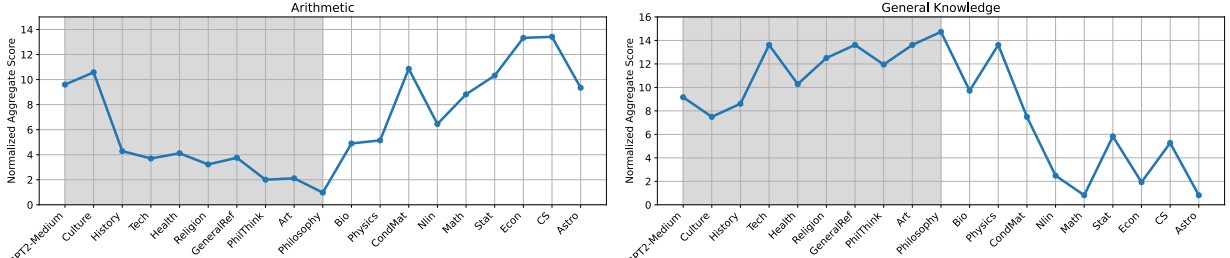

Figure 9: GPT2-M performance on two downstream tasks, captured at L1 domain transitions (the higher the better). The initial points represent the baseline performance of GPT2-M. Please see Figure 12 for other tasks.

domains immediately after completing training on a given L1 domain. Each plot in Figure 23 shows the forward transfer performance to the domain stated in the title. Most notably, panels in the first row reflect that pretraining on a handful of domains leads to significantly worse performance compared to zero-shot (the dotted horizontal lines). In contrast, extended pretraining across a variety of domains occasionally leads to positive forward transfer (panels 2 and 3). Further, we notice a *recency effect* where the forward transfer perplexity improves if a checkpoint is transferred to a domain that is conceptually similar to the most recent training domain: as anticipated, the most successful forward transfer to Astrophysics domains is attained after training on Physics.

## 4.5 Does continual learning influence downstream performance?

Until now, our evaluation has centered on assessing the language modeling capabilities of our models, specifically using perplexity as our performance metric. While training and evaluation performance metrics in CL have traditionally been the same (e.g., classification accuracy), LLMs are often benchmarked on downstream tasks whose metrics (e.g., question-answering accuracy) often differ from the pretraining objective (e.g., perplexity). Likewise, moving forward, we assess the performance of our continually trained models on different tasks. Overall, our findings reveal that continuing pretraining on domains relevant to these tasks generally enhances model performance, while pretraining on unrelated domains often leads to forgetting, thereby negatively affecting the model's initial task proficiency.

We choose five tasks aligned with our benchmark domains from BIG-Bench (bench authors, 2023): Arithmetic, General Knowledge, Physics, CS Algorithms, and Few-shot Natural Language Generation (NLG). We report *Normalized Aggregate Score*, that is, the normalized preferred metric averaged over all subtasks under that particular task (e.g., Arithmetic task has twenty subtasks). In (bench authors, 2023), they specify that the best-performing language models achieved a score below 20, and model scores can be less than zero on some tasks. Comprehensive task descriptions, metrics, and additional outcomes are provided in the Appendix A.3.

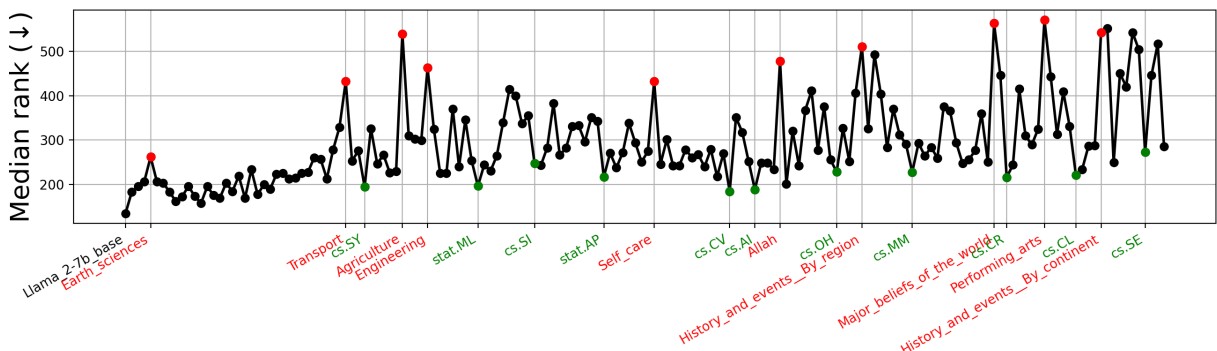

Figure 10: Our rank-based knowledge transfer analysis, where continually pretrained Llama2-7B checkpoints are tested on `cs.AI` domain. The red and green dots highlight some domains on which the rank is significantly high and low. We see an increasing trend, implying a decrease in performance. We also notice that domains that are semantically similar/distant to `cs.AI` consistently lead to better/worse rank.

**GPT2-M downstream performance is determined by continual pretraining domains**   As depicted in Figure 9, a consistent decrease in Arithmetic task performance was noted when models were continually trained on Wiki domains which then improves upon switching to S2ORC domains, with the exception of the Nonlinear Sciences and Astrophysics domains. The performance trends for the CS Algorithms and Physics tasks align with those observed for the Arithmetic task, with peaks reached after training on the CS and Physics domains, respectively (see Figure 12 for the results). In contrast, performance on General Knowledge tasks improved with Wiki domain training but went below the base performance led by training on S2ORC. Overall, we conclude that the performance gain/degradation measured by perplexity in the context of CL transfers to downstream performance.

**Continual Llama2-7B checkpoints consistently achieve chance-level performance**   Next, we repeat the same analysis with Llama2-7B checkpoints. As shown in Figure 13, performance drops drastically, often falling below GPT-1, regardless of the task or training order. Notably, the decline in downstream performance after the first task is catastrophic, starkly contrasting with our findings for GPT2-M. Further analysis reveals that continual pretraining severely degrades the model's ability to perform in-context learning and generate coherent dialogues. This aligns with observations from (bench authors, 2023), which highlight the brittleness of language models—their sensitivity to natural language phrasing. Taken together, our results on continual Llama2-7B checkpoints do not provide immediate insight into the extent of forgetting.

## 4.6   Prediction rank-based analysis for knowledge accumulation

Next, we address the limitations of downstream task evaluation presented in the previous section, which stems from our continually pretrained Llama2-7B checkpoints' inability to be zero/one/few shot prompted. In particular, our rank-based method provides an alternative metric for assessing domain knowledge retention, independent of prompting ability. We also demonstrate that the ranks achieved by Llama2-7B show a much more interpretable pattern than the downstream task performance.

Our proposed pipeline builds on Geva et al. (2023), starting with text corpus preprocessing, including citation removal, LaTeX markup cleaning, and special character filtering. Next, we embed all sentences using Sentence-BERT (Reimers, 2019) and cluster the embeddings. Each cluster center is then associated with a representative *keyword* - an n-gram extracted from the sentences within the cluster. For example, "prior distribution" and "multi-layer perceptron" are keywords related to the `cs.AI` corpus. For each cluster, we extract *target words* - frequently occurring terms with high semantic similarity to the keyword (e.g., "activation" and "forward pass" for "multi-layer perceptron"). We then retrieve sentences from the test corpus containing these keywords. Our continually pretrained LLMs process these sentences up to the keyword, where the keyword itself serves as the ground truth prediction target. Our key metric is the rank of the target token, which reflects the models' ability to complete domain-specific sentences. By focusing on domain and subject-specific contexts, our approach quantifies knowledge accumulation in a text corpus without being influenced

by prompt variations. We remind that our pipeline is domain invariant and can be applied to any text corpus. This is particularly useful for niche domains (e.g., `Cosmology and Nongalactic Astrophysics`) for which there may be no corresponding downstream task.

Figure 10 presents the average rank of a Llama2-7B model continually pretrained on randomly ordered domains, checkpointed after each L2 domain and evaluated across all target words associated with all keywords in the `cs.AI` domain (see Figure 14 for a Llama2-7B model trained in a similar order). Strikingly, the original Llama2-7B model achieves the lowest rank, i.e., the highest performance. Over the course of continual pretraining, we observe a clear upward trend in rank, indicating a decline in prediction quality with further training. Knowledge transfer also follows a distinct pattern: semantically similar domains, such as `stat.ML` and `cs.CV`, consistently achieve better ranks (signifying positive transfer), whereas more distant domains, such as `Transport` and `Self care`, lead to significantly worse transfer. Figure 15 shows the findings of the same rank analysis on a GPT2-M model trained randomly. We notice an overall rank improvement, hinting at knowledge accumulation. Further, the training domains that achieve the best ranks are all CS domains; therefore, we conclude that our rank-based analysis is somewhat informative on the model's knowledge on a particular topic. These findings align with our downstream analysis on GPT2-M, reinforcing that training domains strongly influence test performance.

## 4.7 Additional observations and ablations

**RoBERTa results differ substantially from decoder-only results**    To broaden our analysis and gain deeper insights into the behavior of different architectures, we have repeated all the experiments with RoBERTa and obtained somewhat surprising results. As shown in Figure 17, RoBERTa models do not seem to exhibit forgetting on old tasks, a finding that aligns with Cossu et al. (2024). This is also visible in Figure 24, where backward transfer perplexity remains similar to continual pretraining perplexity. Finally we observe positive forward transfer most of the time, which also conflicts with the decoder-only model findings. Please see Section A.2 for more detailed analyses.

**LLMs forget more in the later stages of continual learning**    To complement the forgetting findings presented in Section 4.2, we analyze how the forgetting dynamics evolve during continual learning. We divide the checkpoints in random-order training into two groups based on their recency (checkpoints[50-100] and checkpoints[100-150]). We evaluate all checkpoints on the last 50 domains and compute the perplexity change (caused by additional training). Histograms in Figure 8 show that earlier checkpoints exhibit less forgetting, i.e., transfer better to the past. We informally hypothesize that in the earlier stages of training, the parameters that are not *important* to the recently learned tasks are updated and once the models fill *their learning capacity* in the later stages, parameter updates become detrimental.

**Batch size impacts learning dynamics**    As an ablation study, we increase the batch size from 16 to 64, thereby performing a quarter of gradient updates. Figure 20 compares the results obtained with different batch sizes. When trained in random-order, continual pretraining and last checkpoint perplexities virtually remain the same despite varying the batch size. In similar-order, a smaller batch size helps to improve continual pretraining perplexity but worsens the performance of the last checkpoint. We hypothesize that taking more gradient steps aids the model to better fit the current task while promoting forgetting the old tasks.

**Balancing the data size across L2 domains does not improve performance**    We investigate whether the imbalance in training data sizes impacts the overall performance (see Table 1 for L1 domain lengths). To address this, we set the number of maximum tokens to 100K for each L2 domains (if they have fewer tokens, we used them all), and train the original model. Figure 21 shows the resulting continual pretraining and last checkpoint perplexities. For both metrics, test performance on almost all L2 domains deteriorates after balancing the number of data points per domain. The results suggest using all data at hand instead of leaving some out for the sake of balanced training.

**Swapping Wiki and S2ORC portions verifies previous findings**    We swap the portions for similar-order training, i.e., training first on S2ORC, then on the Wiki portion. Arguably, this training order still follows conceptual similarity; hence, it allows us to see whether our previous findings still hold. The left panel in Figure 22 shows that continual pretraining perplexity remains almost the same. Yet, the last checkpoint perplexity significantly changes: while the performance on the S2ORC portion substantially degrades, we observe the opposite effect for the Wiki portion. Agreeing with our previous findings, we conclude that the checkpoints perform worse when tested on older domains/portion.

**Alternative random orders yield similar findings** In our random-order experiments, we consider only one randomized training sequence. To test whether the findings do not generalize to alternative randomized orders, we re-shuffle the dataset twice and repeat the experiments with GPT2-M. These experiments resulted in median CPTs of 16.4 and 16.78 while 16.69 is obtained in our main set of experiments. Given relatively much larger differences across different experiment setups, we conjecture that the standard deviation resulting from different random orders can be safely ignored.

## 5 Related work

We discuss two related but separate lines of research in the context of CL for LLMs: (i) continual fine-tuning, which aims at fine-tuning LLMs on a series of downstream tasks, and (ii) continual domain-adaptive pretraining, focusing on incremental updates to adapt an LLM to new domains without exhaustive retraining from scratch upon new data.

**Continual fine-tuning** A large body of CL works for LLMs tries to mitigate forgetting during continual fine-tuning. (Luo et al., 2023a) investigate forgetting and distribution drift during continual learning on a series of eight downstream classification tasks. In a recent work, (Luo et al., 2023b) examines evolution of forgetting during continual fine-tuning. Scialom et al. (2022) instruct fine-tune an LLM for eight tasks. Khan et al. (2022) introduce an adapter-based fine-tuning strategy for three downstream tasks. Zhang et al. (2022) propose to add new modules to a sequence generator (such as an LLM) to continually adapt to five tasks. Razdaibiedina et al. (2023) introduce progressive prompts, where a growing number of prompts, are learned during continual learning, fine-tunes on 15 classification datasets. Wang et al. (2023) propose to learn orthogonal adapters to minimize interference between 15 classification tasks. Qin et al. (2022) propose efficient lifelong pretraining for emerging data (ELLE), where they expand a network during learning and include domain-identifying prompts during pretraining to help the PLM identify the type of knowledge it is learning.

**Continual domain-adaptive pretraining** An alternative research direction, closer to our work, aims to continually pretrain LLMs to adapt them to new domains. In one of the earliest studies, Gururangan et al. (2020) introduce a growing mixture of expert architecture for domain-adaptive continual pretraining and Lazaridou et al. (2021) presents a dataset to assess temporal generalization of language models when trained continuously. Jin et al. (2021) continually pretrain RoBERTa-base over a domain-incremental research paper stream and a chronologically-ordered tweet stream with different continual learning algorithms. In a similar study, Fernandez et al. (2024) proposes layer-specific pretraining and learning rates to mitigate forgetting when a GPT2-L model is adapted to (temporally changing) Wikipedia snapshots. Chen et al. (2023) study lifelong learning from a sequence of online pretraining corpus distributions based on a progressively growing mixture-of-experts (MoE) architecture. Likewise, Gururangan et al. (2021) introduce a mixture architecture for continual adaptation. Ke et al. (2023a) show how a soft-masking mechanism for gradients of RoBERTa model could be useful for domain-adaptive pretraining for eight tasks. Cossu et al. (2024) investigate the characteristics of the continual pretraining across ten domains. Duwal et al. (2024); Vo et al. (2024) study domain adaptation of Llama3-8B model to a new language (Nepali) using quantized low-rank adaptation. The more elaborate study of Gogoulou et al. (2024) includes morphologically similar three (North European) languages and models up to 1.3B parameters. A very interesting technical report by Vo et al. (2024) shows that combining diverse pretraining stages (an initial training of the embedding layers, pretraining of the entire network, and finally low-rank adaptation) yields impressive performance on Korean language. Ji et al. (2024) continually pretrain a Llama2-7B model on 250k question-answer pairs but do not investigate forgetting. Chen et al. (2024) discover that LLMs suffer from forgetting in "continual memorization" tasks and replay methods cannot fully resolve the problem. Gupta et al. (2023) examine different warm-up strategies for continual pretraining. Öncel et al. (2024) studies when and why domain-adaptive pretraining of LLMs may fail. Finally, Fisch et al. (2023) introduce a benchmark of task sequences that potentially lead to positive and negative transfer and further propose a simple strategy for robust forward transfer, which aims to pick the checkpoint with the biggest positive knowledge transfer among all past task checkpoints. Our work diverges from the others in that we continually pretrain the original model without any expansion on a much longer horizon of 159 domains, and further investigate the impact of domain order.

# 6 Discussion

Prior studies in CL for LLMs have mainly focused on parameter-efficient fine-tuning or adaptation for a limited selection of target domains or tasks. While beneficial, these methods often do not fully address the broader challenge of lifelong learning for LLMs. Our research diverges by exploring continual domain-adaptive pretaining of LLMs across an extensive set of domains to better understand the dynamics of knowledge preservation, new information retention and knowledge transfer. Below, we highlight three key insights and discuss notable observations from our research:

**Continual pretraining benefits GPT models but degrades Llama2-7B** While continual pretraining improves GPT models of all sizes, it negatively impacts Llama2-7B. This trend is consistent across all our metrics - perplexity, downstream task performance, and output rank. Regardless of model size, continual pretraining outperforms domain-adaptive pretraining, underscoring the advantages of exhaustive training. Finally, we empirically show that Llama2-7B requires domains to be larger than 100 MB for improved adaptation.

**Scaling laws for continual pretraining** Our experiments reveal that larger models consistently achieve better perplexity and exhibit less forgetting. However, smaller models are more sensitive to continual pretraining, showing both the highest degree of learning and the most pronounced forgetting.

**Continual pretraining boosts downstream task performance of GPT family** For both GPT2-M and Llama2-7B, performance on BIG-Bench tasks follows the same trend as perplexity. Further, downstream task success is heavily influenced by the most recent training domains and correlates well with our complementary prediction rank analysis. A key takeaway is that additional generative pretraining before supervised fine-tuning can further enhance performance.

**Randomizing training domain order improves knowledge retention** With the randomized training order, we observe two benefits: that *(i)* the final checkpoint achieves superior average performance compared to similar-order training, and *(ii)* checkpoints exhibit positive backward transfer and reduced forgetting, suggesting that previously learned knowledge remains more intact.

**Similar-order training strengthens domain specialization** When consecutive training domains are semantically similar, continual learning enhances specialization. This is supported by two observations: (i) continual pretraining achieves lower perplexity in similar-order training, likely due to gradual knowledge accumulation, and (ii) models exhibit positive transfer to recent past domains but not to more distant ones in the training sequence.

# 7 Limitations

Our research highlights CL as a powerful paradigm for learning in LLMs, providing valuable insights into its mechanisms and benefits. However, we acknowledge several limitations in our study:

- Since part of RoBERTa 's training data includes Wikipedia entries, potential overlap with our training set may have influenced its results.

- To report average backward transfer perplexity, we exhaustively evaluated all checkpoints across all past domains, totaling 12,561 evaluations per model per setup. Forward transfer was assessed only after completing all L2 domains within a given L1 domain, still requiring 171 evaluations per model per setup.

- Computation time remains a fundamental limitation. With experiments spanning 159 tasks, the sequential nature of continual pretraining prevents parallelization of the training process.

- Investigating how the original pretraining corpus (e.g., WebText for GPT-2) impacts continual learning of individual domains would be an interesting future work.

**Acknowledgements** Çağatay Yıldız and Matthias Bethge are members of the Machine Learning Cluster of Excellence, funded by the Deutsche Forschungsgemeinschaft (DFG, German Research Foundation) under Germany's Excellence Strategy – EXC number 2064/1 – Project number 390727645. This research utilized compute resources at the Tübingen Machine Learning Cloud, DFG FKZ INST 37/1057-1 FUGG!

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

# A Appendix

## A.1 Key distinguishing aspects

We hope that our research marks a shift towards establishing a more realistic benchmark for investigating CL in LLMs and hope that our interesting findings lay the groundwork for future studies in this realm. With this, we would like to highlight some key distinguishing aspects of our work.

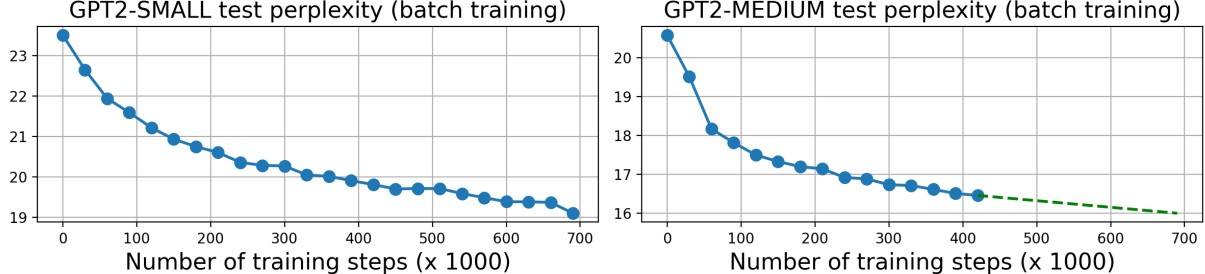

Figure 11: We mix all the samples in all the domains and train GPT2-S and GPT2-M models on the mixed dataset. One epoch corresponds to 700k iterations. We observe that the final checkpoint has comparable perplexity (19.2) to continual pretraining perplexities in the similar and random orders (19.38 and 19.67). Importantly, it is significantly larger than domain adaptive pretraining perplexity (21.48) and the final checkpoints of continual pretraining in both orders (24.79 and 20.5). For GPT2-M, due to compute constraints, we had to end the training at 420k iterations. Extrapolating the perplexity curve reveals the same findings as GPT2-S.

- A single continual pretraining run with the smallest (GPT2-S) and largest (Llama2-7B) models in our benchmark required 6 days and 4 months of training, respectively. Further, we compute the perplexity of all checkpoints on all previous domains, which adds up to 12,561 evaluations. This scale stands in stark contrast to conventional benchmarks such as split CIFAR-100 (Krizhevsky et al., 2009) and Tiny ImageNet (Le & Yang, 2015), which operate on a much smaller scale.

- The continual learning tasks in some of the previous works (Cossu et al., 2024) involve end-task fine-tuning to evaluate the performance of the continually trained LLM. On the contrary, most training domains considered in this work are not tied to a particular downstream task.

- Our key findings in Figure 3 highlight perplexity as a primary metric. While this may seem at odds with standard LLM benchmarking practices, it aligns with the continual learning literature, where the primary optimization objective is consistently tracked and reported. Notably, we observe a strong correlation between perplexity and downstream task performance, reinforcing its value as a proxy for measuring knowledge transfer and forgetting.

- Our benchmark facilitates the study of how semantic ordering of training domains impacts performance - an aspect largely unexplored in previous research.

## A.2 Perplexity in masked language models always increases with continual learning

To broaden our analysis and gain deeper insights into the behavior of different architectures, we have repeated all the experiments with RoBERTa and obtained somewhat counter-intuitive and surprising results. First of all, we want to point out that perplexity is not well-defined for masked language models like RoBERTa[1]. In our experiments, we used the same perplexity computation for RoBERTa as the one we used for GPT2, which is equal to $\exp\{-\frac{1}{t}\sum_i^t \log p_\theta(x_i|x_{<i})\}$ where $X = (x_0, \ldots, x_t)$ is the tokenized sequence and $\log p_\theta(x_i|x_{<i})$ is the log-likelihood of the $i$th token conditioned on the preceding tokens $X_{<i}$ according to the model.

**Forgetting in RoBERTa family is not evident** In contrast to the GPT family, our analysis reveals that the RoBERTa family does not exhibit forgetting of old tasks during continual training. To illustrate this, we present a visualization of backward transfer performance across four randomly selected domains in Figure 24. Similar findings with encoder-decoder models were reported in (Cossu et al., 2024). We conjecture that modifying the model architecture by including a bottleneck layer plays a significant role in this behavior.

---

[1]https://huggingface.co/docs/transformers/perplexity

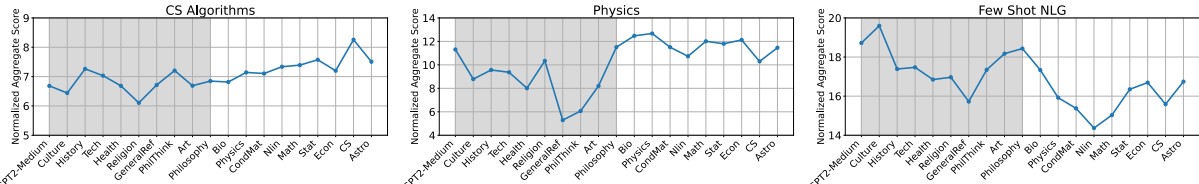

Figure 12: GPT2-M performance on the remaining three downstream tasks from BIG-Bench, captured at L1 domain transitions. The initial point represents the baseline performance of GPT2-M. The CS Algorithms and Physics results align with the findings in the main paper. For the Few-shot NLG task, performance trends across Wiki and S2ORC domains do not follow a consistent pattern. A deeper inspection reveals that domains such as Culture, Art, Philosophy, Math, Stat, and Econ contribute positively to performance enhancement in this task.

**RoBERTa-large always exhibits positive backward transfer while GPT2-L transfer performance depends heavily on the transferred domain** Looking into Figure 24, we notice that backward transfer perplexity of RoBERTa-large remains relatively close to fine-tuning performance. Interestingly, we observe occasional jumps in perplexity when trained in random order, whose analysis is an interesting future work. On the other hand, Figure 19 demonstrates backward transfer to the same four domains when GPT2-L is trained. In agreement with our earlier findings, switching from Wiki portion to S2ORC causes a significant perplexity degradation on Wiki domains when trained in similar order. Further, the characteristics of the test domain seem to determine whether the transfer is positive or negative. Finally, we observe a less fluctuating backward perplexity with random training order.

**Encoder-decoder models require just a few L1 domains for good transfer** *(i)* In stark contrast with the decoder-only models, pretraining even on the first L1 domain helps to exceed zero-shot performance (comparing the dotted lines and the first point of each sequence). Interestingly, this holds when the pretraining and test domains belong to different portions of the training set. *(ii)* We further notice the forward transfer perplexity tends to improve for the first ten L1 domains and later slightly degrade. Since it is still considerably above zero-shot performance, we chose not to investigate this in detail. *(iii)* Lastly, the model size does not seem to influence forward transfer performance, which is again as opposed to decoder-only models.

## A.3   BIG-Bench Experiments

**Tasks.** We selected five tasks that align with our benchmark domains, as described below:

*Arithmetic* evaluates the model's ability in basic arithmetic operations – addition, subtraction, multiplication, and division – ranging from 1-digit to 5-digit numbers.

*General Knowledge* assesses the model's ability to answer questions across a broad spectrum of general knowledge, for example, "How many legs does a horse have?". It draws parallels with benchmarks focused on general-knowledge question-answering, such as those found in (Rajpurkar et al., 2016).

*Physics* aims to test the model's understanding of physics by asking it to determine which formula is needed to solve a given physics word problem, and evaluating the accuracy of the multiple choice responses. The decision to utilize a multiple-choice format concentrates on the model's comprehension of the physical principles each formula represents, addressing concerns that generating physics formulas through text might be overly challenging for current models.

*CS Algorithms* measures the model's performance on two core algorithmic concepts: recursion (or stack usage) and dynamic programming, evaluating the model's computational thinking and problem-solving skills.

*Language Generation from Structured Data and Schema Descriptions (Few-shot NLG)* aims to assess the ability of a model to generate coherent natural language from structured data, supported by schema descriptions, within the framework of a task-oriented dialogue system. The goal is to determine whether a virtual assistant can learn to generate responses based on the textual description of structured data, enabling rapid adaptation to new domains with minimal additional input.

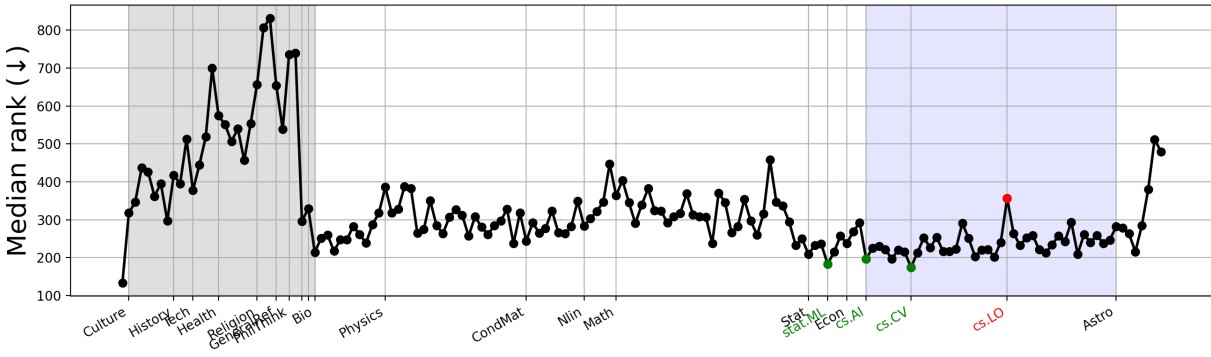

Figure 13: Llama2-7B performance on all downstream tasks from BIG-Bench. The initial data point represents the baseline performance of Llama2-7B. We notice that only after a few L2 domains, model performances immediately drop and never recover the base model.

Figure 14: Our rank analysis for Llama2-7B model trained on similar order. The grey/purple backgrounds show the Wiki and `CS` domains. We notice that the training domains clearly influence the transfer to `cs.AI` domain.

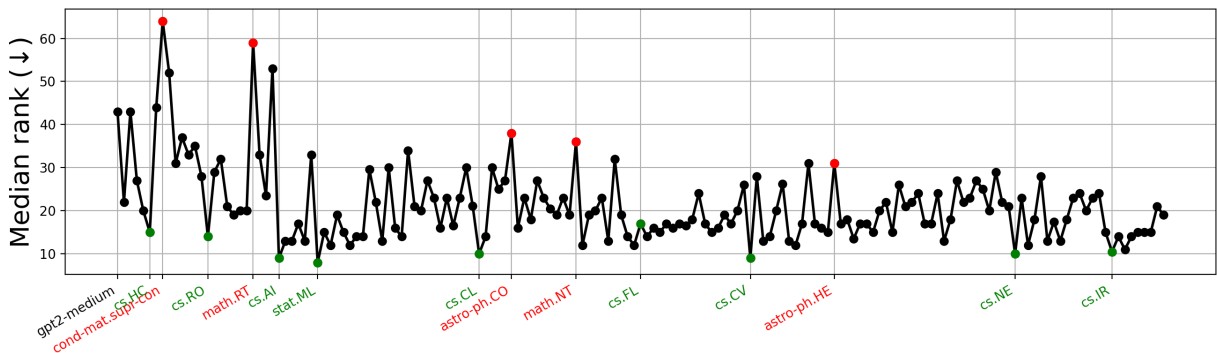

Figure 15: Our rank analysis for GPT2-M. We see that continual pretraining in random order, overall helps the model achieve better rank on `cs.AI` test set. This finding is in agreement with the downstream task analysis presented in Section 4.5.

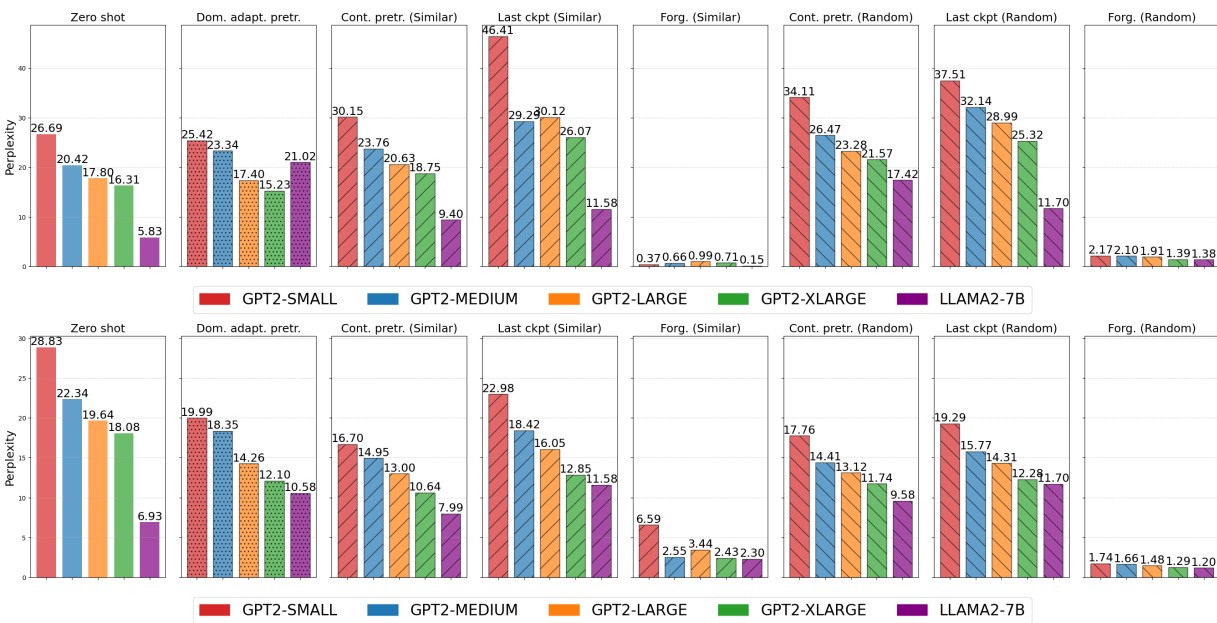

Figure 16: A more detailed analysis of our main results table. This time, we compute the test perplexities($\downarrow$) on Wiki and S2ORC portions separately, visualized in the upper and lower panels.

## A.4 Domain sequences

**How did we order domains?** For random order experiments, we randomly shuffle all L2 domains and use the same domain sequence in all our experiments. For sanity check, we shuffle all L2 domains with different random seeds twice and train GPT2-M on these sequences.

For similar order experiments, we first compute domain similarities by embedding 10K samples from each L1 domain and 50K samples from OpenWebText using Sentence-BERT (Reimers & Gurevych, 2019). Then we set the similarity between two domains as the mean of the cosine similarities between all possible pairs of embeddings from these two domains. Figure 1 shows the obtained similarity matrix. To form our similar order, we start with the Culture domain, which is the most similar to OpenWebText. Subsequent domains are chosen from the unselected ones, where the probability of a domain being selected is proportional to its similarity to the most recently selected domain. We believe this procedure contains some randomness while similarity is ensured thanks to the said probabilistic selection procedure. Again, for sanity check, we consider two more similar orders and train GPT2-S on GPT2-M on them.

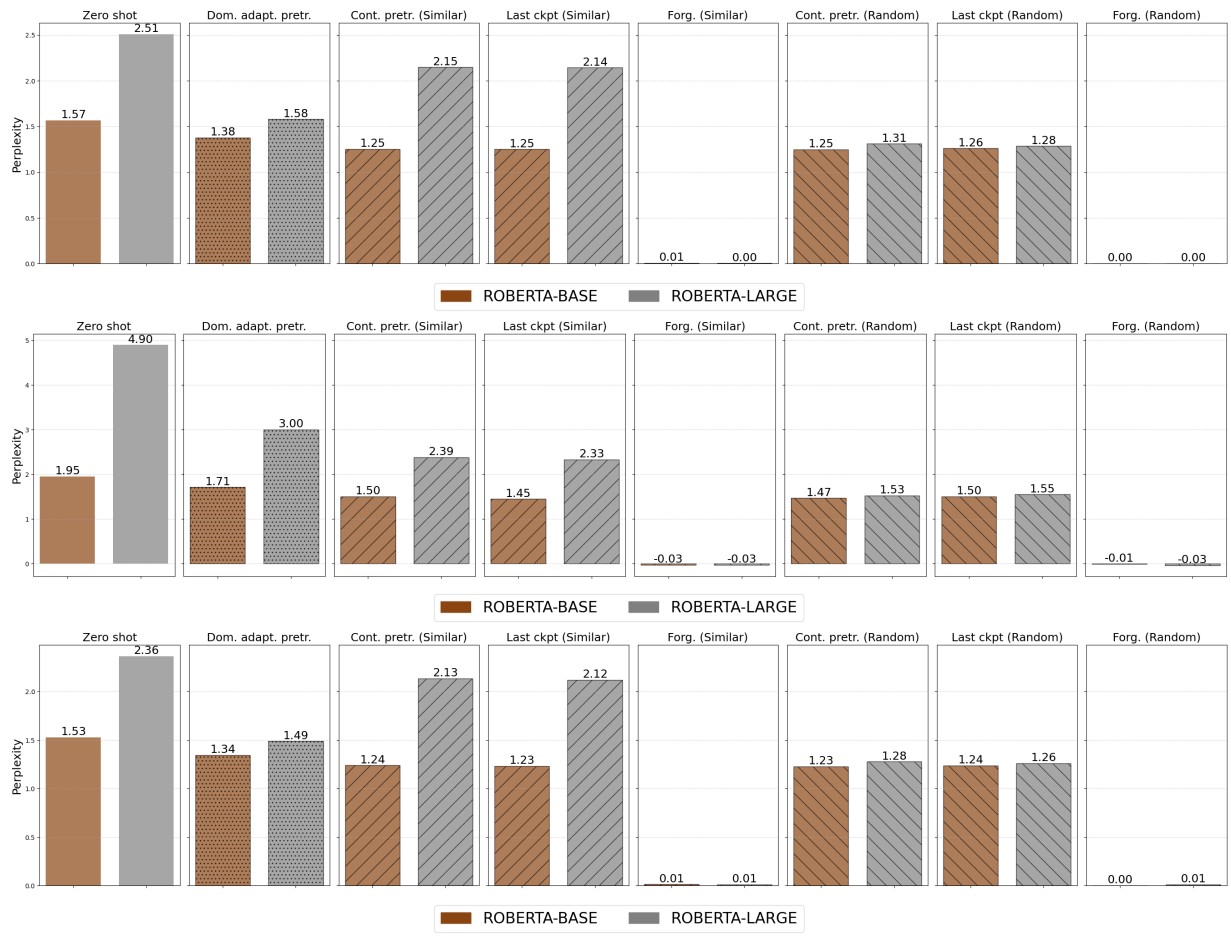

Figure 17: Above plots show the test perplexities (↓) with different RoBERTa sizes and training orders. For reference, we include the zero-shot and fine-tuning perplexities. The first panel shows the cumulative results, and the remaining two show the results obtained on Wiki and S2ORC domains. Inside the parentheses are the perplexity improvements over zero-shot (the smaller the better).

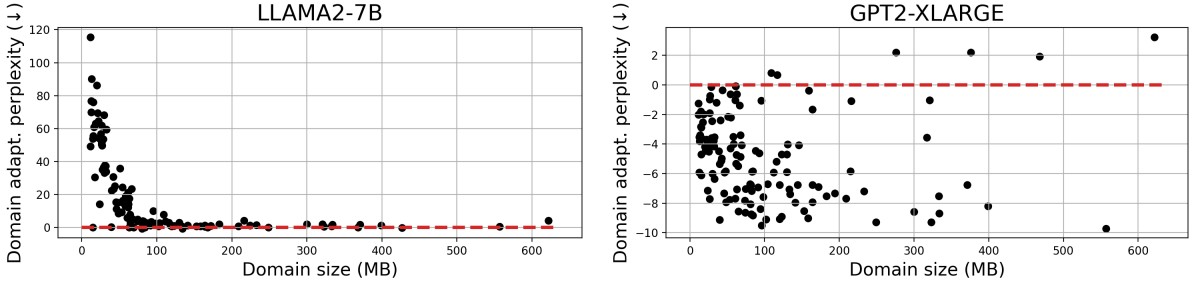

Figure 18: Our analysis of how much domain adaptation improves the perplexity for Llama2-7B and GPT2-XL. Each point corresponds to one L2 domain. The x-axis is the domain sizes and the y-axis is the perplexity after domain adaptation (normalized by zero-shot). Positive y values are the domains on which adaptive pretraining cause a degradation in perplexity.

**Additional similar order experiments**    As noted in the previous paragraph, we train GPT2-S and GPT2-M models on domain sequences that are ordered by semantic similarity. Note that the first domain sequence starts with "Culture" (as in our main experiments) while we choose to start with "astro-ph" in the second domain sequence.

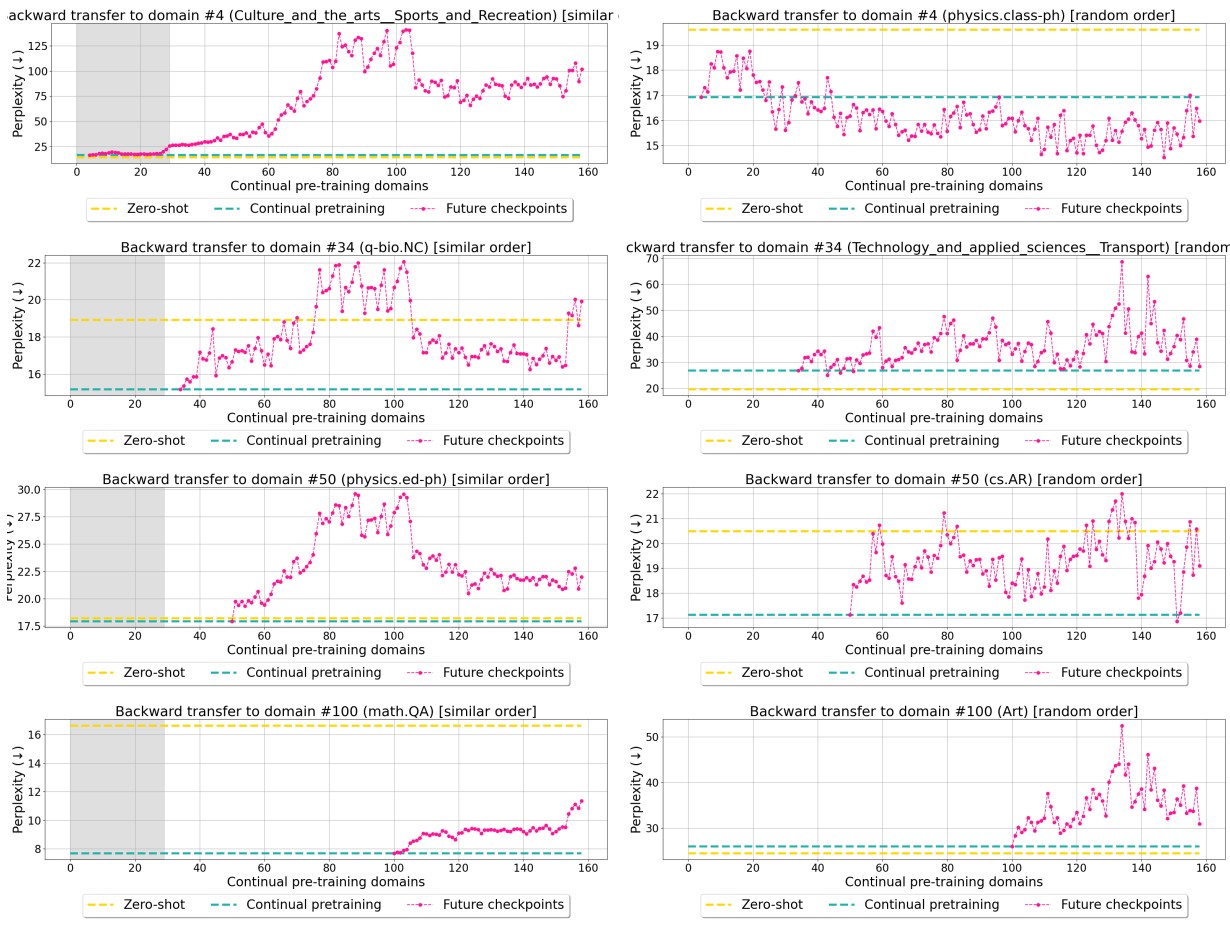

Figure 19: Backward transfer illustration with GPT2-L trained in similar and random order (left and right columns). Each panel shows the backward transfer perplexity (pink) computed on a particular domain as optimization proceeds. For baseline comparisons, we also plot zero-shot (yellow) and continual pretraining (green) perplexities.

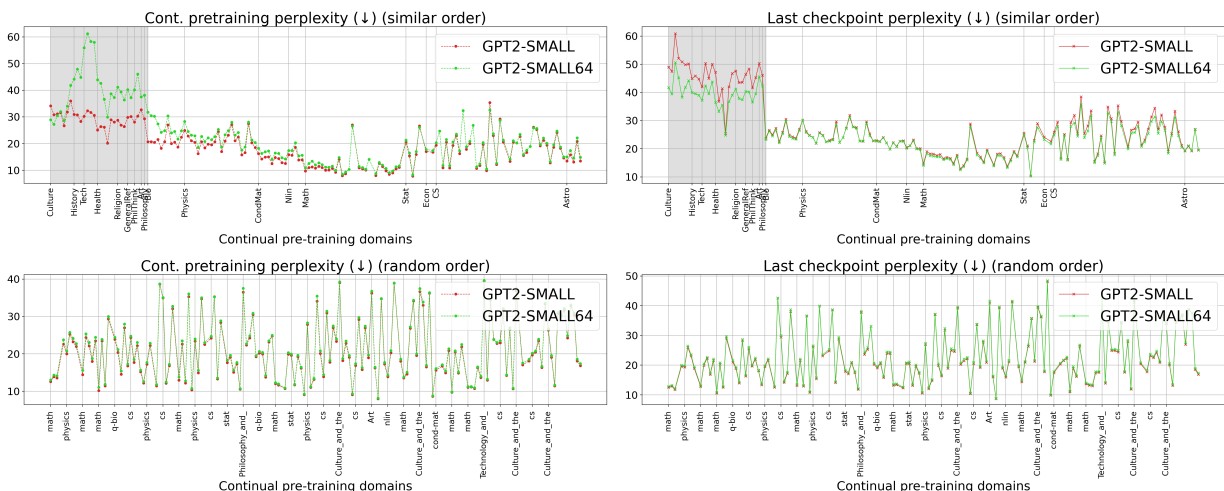

Figure 20: A comparison of GPT2-S training with batch sizes 16 (our default) and 64. For random and similar training orders (rows), we plot the continual pretraining and last checkpoint perplexities (columns).

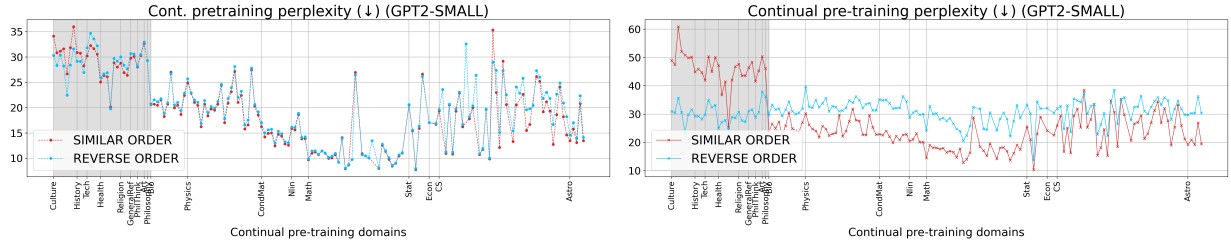

Figure 21: A comparison of GPT2-S training with all available data (our default) as well as a subsample of data with equally many data points per L2 domain. We only train in similar orders and plot the continual pretraining (left) and last checkpoint perplexities (right).

Figure 22: A comparison of GPT2-S training with our default similar training order (Wiki portion, followed by S2ORC) as well as an alternative version (S2ORC portion, followed by Wiki). We plot the continual pretraining and last checkpoint perplexities. Note that the $x$ axis corresponds to the default training order.

The GPT2-S model trained on the new similar order yielded similar continual pretraining (CPT) perplexity (20.25) to what was reported in our main experiments (19.38). Nonetheless, the perplexity of the last checkpoint (38.4) is much worse than the results in the main paper (24.78), which shows that the performance of the final model depends crucially on the domain sequence.

Due to time constraints, we managed to evaluate the first 80 checkpoints of the continually pretrained GPT2-M model. The findings obtained so far align with the takeaways above: CPT (12.51) is quantitatively close to CPT attained on the same domains in our main experiments (13.92). The last checkpoint we evaluated (checkpoint-80) achieved a perplexity of 18.77 on all previous domains while we reported 19.33 in the main paper. We believe these numbers will go up as we train on more domains. In particular, switching from S2ORC to Wiki portion arguably will cause significant degradation on the last checkpoint perplexity since the final model will be most recently trained on Wiki while S2ORC constitutes a larger portion of the test domains. We will include and discuss the results obtained after the completion of the evaluation in our camera-ready.

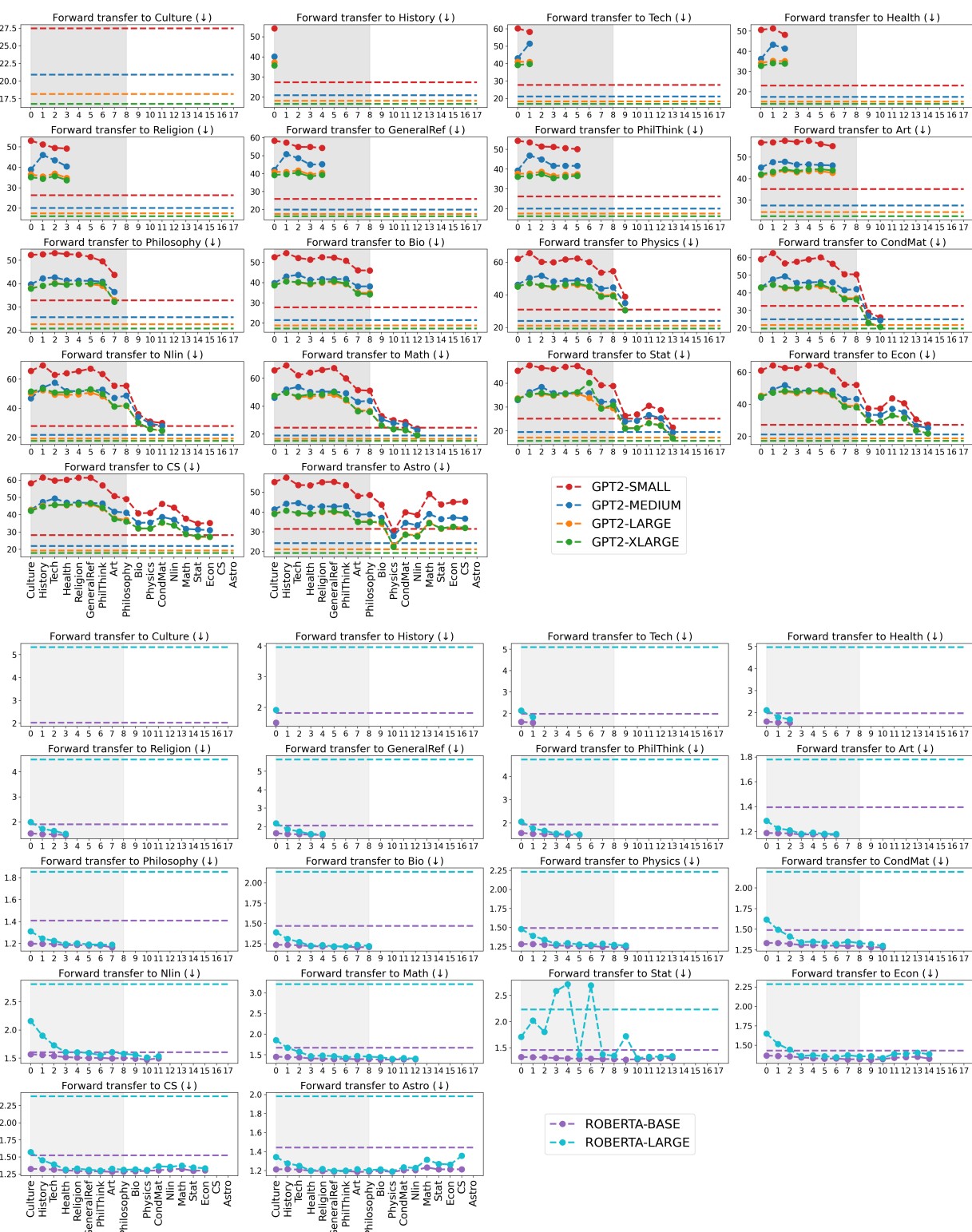

Figure 23: Forward transfer results with similar training order. The checkpoints are saved after having trained on an L1 domain (hence 18 checkpoints per model). The $i$'th panel shows the forward performance on $i$'th domain, obtained by evaluating all previous $i - 1$ checkpoints on that domain. The dashed lines show zero-shot performance. $x$ and $y$ axes correspond to L1 domain names and perplexities, respectively.

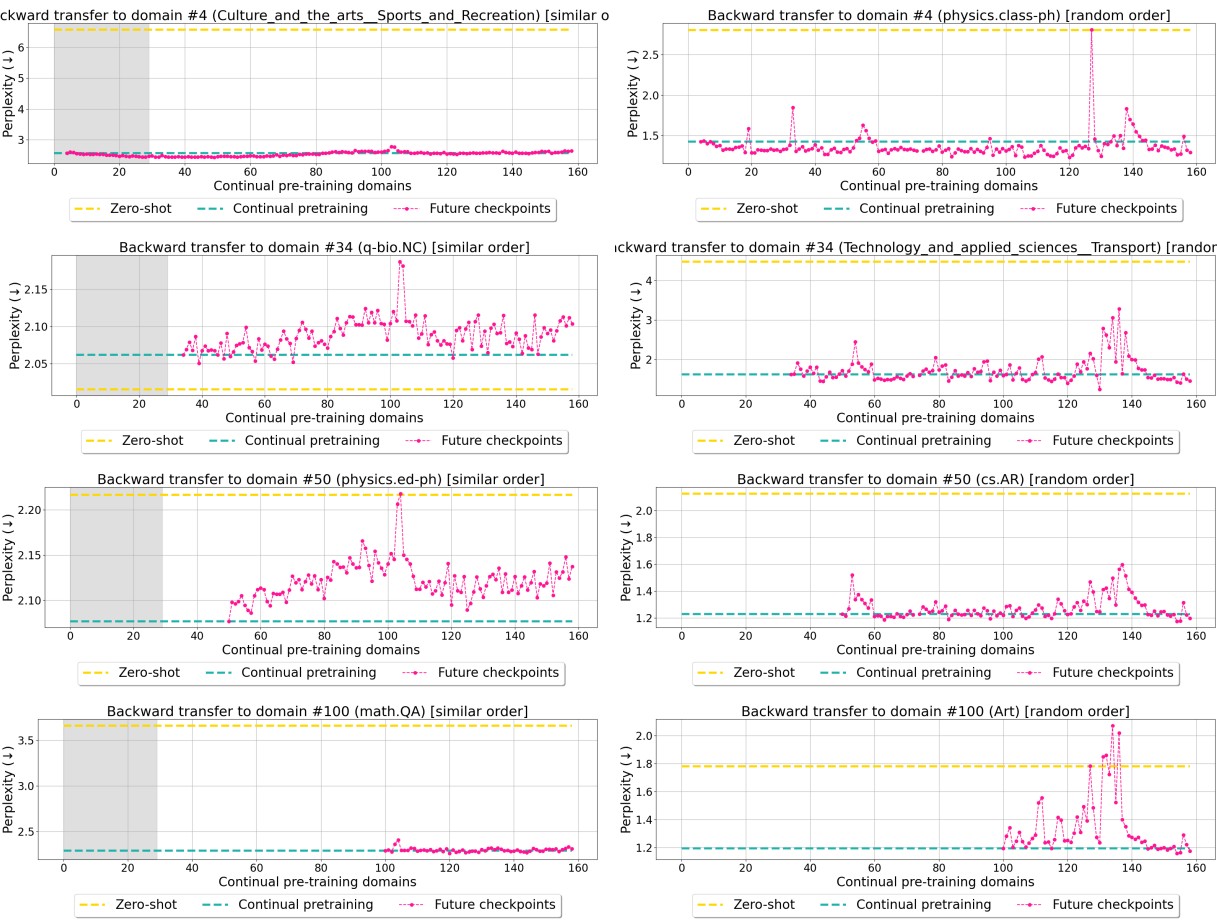

Figure 24: Backward transfer illustration with RoBERTa-large trained in similar and random order (left and right columns). Each panel shows the backward transfer perplexity (pink) computed on a particular domain. For baseline comparisons, we also plot zero-shot (yellow) and continual pretraining (black) perplexities.

Table 2: Order of the domains in our experiments

| Setup | Domains |
|---|---|
| Similar order in our main experiments | "Culture_and_the_arts__Culture_and_Humanities", "Culture_and_the_arts__Games_and_Toys", "Culture_and_the_arts__Mass_media", "Culture_and_the_arts__Performing_arts", "Culture_and_the_arts__Sports_and_Recreation", "Culture_and_the_arts__The_arts_and_Entertainment", "Culture_and_the_arts__Visual_arts", "History_and_events__By_continent", "History_and_events__By_period", "History_and_events__By_region", "Technology_and_applied_sciences__Agriculture", "Technology_and_applied_sciences__Computing", "Technology_and_applied_sciences__Engineering", "Technology_and_applied_sciences__Transport", "Health_and_fitness__Exercise", "Health_and_fitness__Health_science", "Health_and_fitness__Human_medicine", "Health_and_fitness__Nutrition", "Health_and_fitness__Public_health", "Health_and_fitness__Self_care", "Religion_and_belief_systems__Allah", "Religion_and_belief_systems__Belief_systems", "Religion_and_belief_systems__Major_beliefs_of_the_world", "General_referece__Further_research_tools_and_topics", "General_referece__Reference_works", "Philosophy_and_thinking__Philosophy", "Philosophy_and_thinking__Thinking", "Art", "Philosophy", "q-bio", "q-bio.BM", "q-bio.CB", "q-bio.GN", "q-bio.MN", "q-bio.NC", "q-bio.OT", "q-bio.PE", "q-bio.QM", "q-bio.SC", "q-bio.TO", "physics.acc-ph", "physics.ao-ph", "physics.app-ph", "physics.atm-clus", "physics.atom-ph", "physics.bio-ph", "physics.chem-ph", "physics.class-ph", "physics.comp-ph", "physics.data-an", "physics.ed-ph", "physics.flu-dyn", "physics.gen-ph", "physics.geo-ph", "physics.hist-ph", "physics.ins-det", "physics.med-ph", "physics.optics", "physics.plasm-ph", "physics.pop-ph", "physics.soc-ph", "physics.space-ph", "cond-mat.dis-nn", "cond-mat.mes-hall", "cond-mat.mtrl-sci", "cond-mat.other", "cond-mat.quant-gas", "cond-mat.soft", "cond-mat.stat-mech", "cond-mat.str-el", "cond-mat.supr-con", "nlin.AO", "nlin.CD", "nlin.CG", "nlin.PS", "nlin.SI", "math.AC", "math.AG", "math.AP", "math.AT", "math.CA", "math.CT", "math.CV", "math.DG", "math.DS", "math.FA", "math.GM", "math.GN", "math.GR", "math.GT", "math.HO", "math.IT", "math.KT", "math.LO", "math.MG", "math.NA", "math.NT", "math.OA", "math.OC", "math.PR", "math.QA", "math.RA", "math.RT", "math.SG", "math.SP", "math.ST", "stat.AP", "stat.CO", "stat.ME", "stat.ML", "stat.OT", "stat.TH", "econ.EM", "econ.GN", "econ.TH", "cs.AI", "cs.AR", "cs.CC", "cs.CE", "cs.CG", "cs.CL", "cs.CR", "cs.CV", "cs.CY", "cs.DB", "cs.DC", "cs.DL", "cs.DM", "cs.DS", "cs.ET", "cs.FL", "cs.GL", "cs.GR", "cs.GT", "cs.HC", "cs.IR", "cs.IT", "cs.LO", "cs.MA", "cs.MM", "cs.MS", "cs.NA", "cs.NE", "cs.NI", "cs.OH", "cs.OS", "cs.PF", "cs.PL", "cs.RO", "cs.SC", "cs.SD", "cs.SE", "cs.SI", "cs.SY", "astro-ph.CO", "astro-ph.EP", "astro-ph.HE", "astro-ph.IM", "astro-ph.SR" |

| Setup | Domains |
|---|---|
| Random order in main experiments | "math.DS", "cs.CC", "math.MG", "Natural_and_physical_sciences/Earth_sciences", "physics.class-ph", "physics.flu-dyn", "physics.ed-ph", "cs.RO", "q-bio.PE", "stat.TH", "math.DG", "q-bio.BM", "physics.space-ph", "stat.CO", "cs.PF", "math.OA", "physics.bio-ph", "math.GT", "cs.HC", "Mathematics_and_logic/Logic", "q-bio.GN", "nlin.CG", "math.OC", "physics.acc-ph", "nlin.PS", "cs.PL", "cond-mat.mtrl-sci", "physics.gen-ph", "cond-mat.supr-con", "math.LO", "physics.optics", "cs.SD", "math.NT", "math.QA", "Technology_and_applied_sciences/Transport", "cs.GL", "math.SG", "cs.SY", "Philosophy", "math.ST", "math.SP", "cs.MA", "cs.CG", "Technology_and_applied_sciences/Agriculture", "math.AC", "physics.ins-det", "math.NA", "Technology_and_applied_sciences/Engineering", "physics.ao-ph", "cs.IT", "cs.AR", "General_referece/Reference_works", "cs.DS", "physics.hist-ph", "math.IT", "stat.ML", "q-bio.MN", "astro-ph.CO", "cond-mat.dis-nn", "math.RT", "Philosophy_and_thinking/Thinking", "cs.IR", "cs.GR", "Health_and_fitness/Health_science", "cs.SI", "q-bio.QM", "cs.ET", "cond-mat.quant-gas", "physics.med-ph", "stat.OT", "math.CA", "math.CT", "econ.GN", "math.CV", "physics.atm-clus", "stat.AP", "math.PR", "physics.data-an", "nlin.CD", "math.GR", "physics.pop-ph", "cs.DM", "cs.SC", "Health_and_fitness/Self_care", "q-bio.NC", "cs.LO", "cs.CY", "econ.EM", "q-bio.OT", "physics.geo-ph", "Culture_and_the_arts/The_arts_and_Entertainment", "cs.DB", "q-bio.TO", "cs.NI", "math.RA", "cs.CV", "Health_and_fitness/Exercise", "physics.atom-ph", "cs.OS", "cs.AI", "Art", "nlin.AO", "stat.ME", "Religion_and_belief_systems/Allah", "cs.NE", "nlin.SI", "cs.CE", "Culture_and_the_arts/Visual_arts", "Mathematics_and_logic/Fields_of_mathematics", "physics.comp-ph", "math.GM", "astro-ph.HE", "cs.OH", "Health_and_fitness/Public_health", "physics.app-ph", "Culture_and_the_arts/Culture_and_Humanities", "History_and_events/By_region", "cond-mat.other", "Culture_and_the_arts/Mass_media", "math.GN", "cond-mat.soft", "Natural_and_physical_sciences/Physical_sciences", "physics.plasm-ph", "astro-ph.SR", "cs.MM", "math.FA", "q-bio.CB", "cond-mat.stat-mech", "astro-ph.IM", "Mathematics_and_logic/Mathematics", "math.AG", "math.AP", "math.AT", "cs.NA", "cond-mat.str-el", "Technology_and_applied_sciences/Computing", "cs.GT", "Religion_and_belief_systems/Major_beliefs_of_the_world", "Health_and_fitness/Nutrition", "cs.CR", "cs.MS", "Health_and_fitness/Human_medicine", "cond-mat.mes-hall", "cs.DL", "cs.FL", "Culture_and_the_arts/Performing_arts", "Natural_and_physical_sciences/Nature", "physics.chem-ph", "Natural_and_physical_sciences/Biology", "q-bio.SC", "cs.CL", "cs.DC", "math.HO", "astro-ph.EP", "History_and_events/By_continent", "Culture_and_the_arts/Sports_and_Recreation", "physics.soc-ph", "math.KT", "Philosophy_and_thinking/Philosophy", "Religion_and_belief_systems/Belief_systems", "History_and_events/By_period", "cs.SE", "General_referece/Further_research_tools_and_topics", "Culture_and_the_arts/Games_and_Toys", "q-bio", "econ.TH" |

| Setup | Domains |
|---|---|
| Alternative random order used in our ablation | "cs.CR", "math.MG", "Mathematics_and_logic__Logic", "cs.SI", "cs.HC", "cond-mat.mtrl-sci", "cond-mat.supr-con", "math.AP", "physics.med-ph", "nlin.CD", "Culture_and_the_arts__Culture_and_Humanities", "q-bio.CB", "cs.GL", "cs.RO", "cond-mat.other", "math.GN", "Philosophy_and_thinking__Thinking", "stat.OT", "cs.AR", "q-bio.OT", "math.RT", "physics.ed-ph", "q-bio.GN", "cond-mat.mes-hall", "cs.AI", "cs.PF", "econ.TH", "cs.CC", "q-bio.NC", "nlin.SI", "stat.ML", "Health_and_fitness__Self_care", "physics.data-an", "physics.flu-dyn", "Culture_and_the_arts__The_arts_and_Entertainment", "cs.ET", "physics.soc-ph", "cs.NA", "cond-mat.quant-gas", "Health_and_fitness__Health_science", "cs.SY", "math.AG", "cs.CG", "cs.CE", "math.KT", "Health_and_fitness__Human_medicine", "math.AT", "math.DG", "General_referece__Further_research_tools_and_topics", "econ.EM", "math.CA", "cs.DL", "Health_and_fitness__Exercise", "physics.plasm-ph", "Technology_and_applied_sciences__Computing", "cs.CL", "General_referece__Reference_works", "math.SG", "physics.atm-clus", "astro-ph.EP", "astro-ph.CO", "math.OC", "math.LO", "physics.geo-ph", "math.GR", "physics.gen-ph", "Philosophy", "physics.bio-ph", "Health_and_fitness__Nutrition", "Technology_and_applied_sciences__Engineering", "math.NT", "cs.MM", "physics.ao-ph", "Culture_and_the_arts__Games_and_Toys", "Culture_and_the_arts__Performing_arts", "cs.GR", "physics.atom-ph", "cond-mat.dis-nn", "stat.AP", "cs.CY", "cs.FL", "Mathematics_and_logic__Mathematics", "math.IT", "Natural_and_physical_sciences__Biology", "cs.DM", "cs.PL", "Mathematics_and_logic__Fields_of_mathematics", "stat.ME", "q-bio.MN", "History_and_events__By_continent", "Culture_and_the_arts__Visual_arts", "cs.GT", "cs.IT", "math.NA", "q-bio.TO", "math.PR", "nlin.PS", "cs.CV", "cond-mat.str-el", "econ.GN", "physics.comp-ph", "math.RA", "math.CV", "q-bio", "q-bio.QM", "Culture_and_the_arts__Mass_media", "math.QA", "Natural_and_physical_sciences__Physical_sciences", "Health_and_fitness__Public_health", "cs.DS", "astro-ph.HE", "cs.MS", "Natural_and_physical_sciences__Nature", "cs.DC", "math.GM", "physics.pop-ph", "Philosophy_and_thinking__Philosophy", "cond-mat.stat-mech", "q-bio.SC", "cs.OS", "Culture_and_the_arts__Sports_and_Recreation", "math.FA", "math.CT", "History_and_events__By_period", "cs.SC", "math.ST", "physics.space-ph", "cs.SE", "math.DS", "Technology_and_applied_sciences__Transport", "math.AC", "physics.ins-det", "cond-mat.soft", "physics.hist-ph", "math.HO", "math.GT", "math.SP", "Art", "cs.NE", "math.OA", "cs.SD", "Religion_and_belief_systems__Major_beliefs_of_the_world", "astro-ph.SR", "cs.DB", "Technology_and_applied_sciences__Agriculture", "cs.OH", "astro-ph.IM", "Religion_and_belief_systems__Allah", "physics.optics", "physics.app-ph", "Natural_and_physical_sciences__Earth_sciences", "physics.acc-ph", "stat.CO", "cs.IR", "nlin.CG", "cs.MA", "cs.LO", "q-bio.BM", "q-bio.PE", "stat.TH", "physics.class-ph", "physics.chem-ph", "History_and_events__By_region", "Religion_and_belief_systems__Belief_systems", "cs.NI", "nlin.AO" |

| Setup | Domains |
|---|---|
| Alternative random order-2 used in our ablation | "cs.SD", "q-bio.QM", "physics.acc-ph", "cs.DS", "Art", "econ.TH", "math.CA", "math.SP", "Technology_and_applied_sciences__Transport", "cs.MS", "physics.hist-ph", "Natural_and_physical_sciences__Nature", "stat.OT", "math.AG", "General_referece__Further_research_tools_and_topics", "math.LO", "cs.GL", "Technology_and_applied_sciences__Computing", "math.KT", "math.OA", "Culture_and_the_arts__Mass_media", "math.FA", "math.PR", "cs.CG", "physics.ed-ph", "cond-mat.dis-nn", "cs.SE", "Health_and_fitness__Public_health", "Mathematics_and_logic__Logic", "Philosophy", "Mathematics_and_logic__Mathematics", "physics.gen-ph", "stat.ME", "History_and_events__By_continent", "math.CT", "physics.comp-ph", "cond-mat.other", "cond-mat.supr-con", "Culture_and_the_arts__Performing_arts", "physics.class-ph", "physics.app-ph", "q-bio", "Culture_and_the_arts__Visual_arts", "Culture_and_the_arts__Culture_and_Humanities", "cs.CE", "Culture_and_the_arts__Sports_and_Recreation", "nlin.SI", "Mathematics_and_logic__Fields_of_mathematics", "cs.OH", "cond-mat.mtrl-sci", "physics.geo-ph", "physics.pop-ph", "History_and_events__By_region", "astro-ph.EP", "cs.DL", "cs.DM", "cond-mat.stat-mech", "nlin.CD", "math.DG", "math.MG", "cs.GR", "cs.CV", "cs.PL", "cs.CC", "math.NA", "math.QA", "cs.AI", "cs.CL", "physics.space-ph", "stat.AP", "q-bio.SC", "Health_and_fitness__Health_science", "cs.MA", "cs.CY", "cs.ET", "q-bio.CB", "q-bio.GN", "physics.ao-ph", "q-bio.BM", "math.AC", "cs.DB", "cs.DC", "math.HO", "stat.CO", "cs.NE", "math.IT", "q-bio.PE", "econ.GN", "physics.atm-clus", "cs.IT", "Culture_and_the_arts__The_arts_and_Entertainment", "physics.chem-ph", "cs.MM", "cs.HC", "Philosophy_and_thinking__Philosophy", "Health_and_fitness__Human_medicine", "General_referece__Reference_works", "math.RA", "astro-ph.SR", "astro-ph.IM", "physics.bio-ph", "econ.EM", "cs.SC", "math.OC", "physics.soc-ph", "stat.ML", "cs.AR", "physics.med-ph", "cs.FL", "Natural_and_physical_sciences__Earth_sciences", "Religion_and_belief_systems__Allah", "q-bio.TO", "math.SG", "Religion_and_belief_systems__Belief_systems", "Technology_and_applied_sciences__Agriculture", "math.ST", "cond-mat.quant-gas", "math.CV", "physics.atom-ph", "physics.plasm-ph", "Philosophy_and_thinking__Thinking", "nlin.CG", "cs.GT", "physics.data-an", "cond-mat.mes-hall", "cs.SY", "math.NT", "cond-mat.soft", "physics.ins-det", "Natural_and_physical_sciences__Biology", "Technology_and_applied_sciences__Engineering", "physics.flu-dyn", "cs.PF", "math.GR", "q-bio.MN", "cs.RO", "q-bio.NC", "cs.CR", "Health_and_fitness__Self_care", "cs.LO", "cs.SI", "q-bio.OT", "astro-ph.HE", "Religion_and_belief_systems__Major_beliefs_of_the_world", "cs.IR", "cs.NI", "math.DS", "astro-ph.CO", "History_and_events__By_period", "cs.OS", "Culture_and_the_arts__Games_and_Toys", "math.AT", "cond-mat.str-el", "math.GN", "math.GM", "physics.optics", "Health_and_fitness__Nutrition", "nlin.PS", "math.RT", "nlin.AO", "cs.NA", "math.AP", "math.GT", "Health_and_fitness__Exercise", "Natural_and_physical_sciences__Physical_sciences", "stat.TH" } |

| Setup | Domains |
|---|---|
| Alternative similar order (GPT2-S) used in our ablations | { "Culture_and_the_arts__Games_and_Toys", "Culture_and_the_arts__Culture_and_Humanities", "Culture_and_the_arts__Mass_media", "Culture_and_the_arts__Performing_arts", "Culture_and_the_arts__Sports_and_Recreation", "Culture_and_the_arts__The_arts_and_Entertainment", "Culture_and_the_arts__Visual_arts", "History_and_events__By_continent", "General_referece__Further_research_tools_and_topics", "General_referece__Reference_works", "Art", "Philosophy", "Philosophy_and_thinking__Philosophy", "Philosophy_and_thinking__Thinking", "Religion_and_belief_systems__Allah", "Religion_and_belief_systems__Belief_systems", "Religion_and_belief_systems__Major_beliefs_of_the_world", "History_and_events__By_period", "History_and_events__By_region", "Health_and_fitness__Exercise", "Health_and_fitness__Health_science", "Health_and_fitness__Human_medicine", "Health_and_fitness__Nutrition", "Health_and_fitness__Public_health", "Health_and_fitness__Self_care", "Technology_and_applied_sciences__Agriculture", "Technology_and_applied_sciences__Computing", "Technology_and_applied_sciences__Engineering", "Technology_and_applied_sciences__Transport", "cs.AI", "cs.AR", "cs.CC", "cs.CE", "cs.CG", "cs.CL", "cs.CR", "cs.CV", "cs.CY", "cs.DB", "cs.DC", "cs.DL", "cs.DM", "cs.DS", "cs.ET", "cs.FL", "cs.GL", "cs.GR", "cs.GT", "cs.HC", "cs.IR", "cs.IT", "cs.LO", "cs.MA", "cs.MM", "cs.MS", "cs.NA", "cs.NE", "cs.NI", "cs.OH", "cs.OS", "cs.PF", "cs.PL", "cs.RO", "cs.SC", "cs.SD", "cs.SE", "cs.SI", "cs.SY", "stat.AP", "stat.CO", "stat.ME", "stat.ML", "stat.OT", "stat.TH", "econ.EM", "econ.GN", "econ.TH", "math.AC", "math.AG", "math.AP", "math.AT", "math.CA", "math.CT", "math.CV", "math.DG", "math.DS", "math.FA", "math.GM", "math.GN", "math.GR", "math.GT", "math.HO", "math.IT", "math.KT", "math.LO", "math.MG", "math.NA", "math.NT", "math.OA", "math.OC", "math.PR", "math.QA", "math.RA", "math.RT", "math.SG", "math.SP", "math.ST", "physics.acc-ph", "physics.ao-ph", "physics.app-ph", "physics.atm-clus", "physics.atom-ph", "physics.bio-ph", "physics.chem-ph", "physics.class-ph", "physics.comp-ph", "physics.data-an", "physics.ed-ph", "physics.flu-dyn", "physics.gen-ph", "physics.geo-ph", "physics.hist-ph", "physics.ins-det", "physics.med-ph", "physics.optics", "physics.plasm-ph", "physics.pop-ph", "physics.soc-ph", "physics.space-ph", "q-bio", "q-bio.BM", "q-bio.CB", "q-bio.GN", "q-bio.MN", "q-bio.NC", "q-bio.OT", "q-bio.PE", "q-bio.QM", "q-bio.SC", "q-bio.TO", "cond-mat.dis-nn", "cond-mat.mes-hall", "cond-mat.mtrl-sci", "cond-mat.other", "cond-mat.quant-gas", "cond-mat.soft", "cond-mat.stat-mech", "cond-mat.str-el", "cond-mat.supr-con", "nlin.AO", "nlin.CD", "nlin.CG", "nlin.PS", "nlin.SI", "astro-ph.CO", "astro-ph.EP", "astro-ph.HE", "astro-ph.IM", "astro-ph.SR" |

| Setup | Domains |
|---|---|
| Alternative similar order ( GPT2-M ) used in our ablations | "astro-ph.CO", "astro-ph.EP", "astro-ph.HE", "astro-ph.IM", "astro-ph.SR", "physics.acc-ph", "physics.ao-ph", "physics.app-ph", "physics.atm-clus", "physics.atom-ph", "physics.bio-ph", "physics.chem-ph", "physics.class-ph", "physics.comp-ph", "physics.data-an", "physics.ed-ph", "physics.flu-dyn", "physics.gen-ph", "physics.geo-ph", "physics.hist-ph", "physics.ins-det", "physics.med-ph", "physics.optics", "physics.plasm-ph", "physics.pop-ph", "physics.soc-ph", "physics.space-ph", "stat.AP", "stat.CO", "stat.ME", "stat.ML", "stat.OT", "stat.TH", "econ.EM", "econ.GN", "econ.TH", "cond-mat.dis-nn", "cond-mat.mes-hall", "cond-mat.mtrl-sci", "cond-mat.other", "cond-mat.quant-gas", "cond-mat.soft", "cond-mat.stat-mech", "cond-mat.str-el", "cond-mat.supr-con", "math.AC", "math.AG", "math.AP", "math.AT", "math.CA", "math.CT", "math.CV", "math.DG", "math.DS", "math.FA", "math.GM", "math.GN", "math.GR", "math.GT", "math.HO", "math.IT", "math.KT", "math.LO", "math.MG", "math.NA", "math.NT", "math.OA", "math.OC", "math.PR", "math.QA", "math.RA", "math.RT", "math.SG", "math.SP", "math.ST", "nlin.AO", "nlin.CD", "nlin.CG", "nlin.PS", "nlin.SI", "cs.AI", "cs.AR", "cs.CC", "cs.CE", "cs.CG", "cs.CL", "cs.CR", "cs.CV", "cs.CY", "cs.DB", "cs.DC", "cs.DL", "cs.DM", "cs.DS", "cs.ET", "cs.FL", "cs.GL", "cs.GR", "cs.GT", "cs.HC", "cs.IR", "cs.IT", "cs.LO", "cs.MA", "cs.MM", "cs.MS", "cs.NA", "cs.NE", "cs.NI", "cs.OH", "cs.OS", "cs.PF", "cs.PL", "cs.RO", "cs.SC", "cs.SD", "cs.SE", "cs.SI", "cs.SY", "q-bio", "q-bio.BM", "q-bio.CB", "q-bio.GN", "q-bio.MN", "q-bio.NC", "q-bio.OT", "q-bio.PE", "q-bio.QM", "q-bio.SC", "q-bio.TO", "Health_and_fitness__Exercise", "Health_and_fitness__Health_science", "Health_and_fitness__Human_medicine", "Health_and_fitness__Nutrition", "Health_and_fitness__Public_health", "Health_and_fitness__Self_care", "Culture_and_the_arts__Games_and_Toys", "Culture_and_the_arts__Culture_and_Humanities", "Culture_and_the_arts__Mass_media", "Culture_and_the_arts__Performing_arts", "Culture_and_the_arts__Sports_and_Recreation", "Culture_and_the_arts__The_arts_and_Entertainment", "Culture_and_the_arts__Visual_arts", "Art", "General_referece__Further_research_tools_and_topics", "General_referece__Reference_works", "Philosophy_and_thinking__Philosophy", "Philosophy_and_thinking__Thinking", "Philosophy", "History_and_events__By_continent", "History_and_events__By_period", "History_and_events__By_region", "Religion_and_belief_systems__Allah", "Religion_and_belief_systems__Belief_systems", "Religion_and_belief_systems__Major_beliefs_of_the_world", "Technology_and_applied_sciences__Agriculture", "Technology_and_applied_sciences__Computing", "Technology_and_applied_sciences__Engineering", "Technology_and_applied_sciences__Transport" |

## A.5 Learning rate selection

In our experiments, we used a fixed learning rate of 5e-5. We acknowledge that learning rate selection plays a particularly critical role in continual pretraining, where subtle changes in optimization dynamics can significantly affect both knowledge acquisition and forgetting. In our experiments:

- For GPT2 models, we consistently observed that both training and test perplexities decreased, especially in larger domains (>50MB). Higher learning rates sometimes helped smaller domains converge faster, though not always consistently. However, we found that training loss behavior on the current domain did not reliably predict test perplexity behavior on other domains. In other words, sometimes rapid convergence on one domain led to performance degradation on others, and vice versa. These observations suggest that there is no universal or robust learning rate recipe for continual pretraining across heterogeneous domains.

- For Llama2-7B, learning rate sensitivity was even more pronounced. Smaller learning rates often resulted in almost no model updates, while larger learning rates led to both larger improvements and larger degradations, depending on the domain. We settled on 5e-5 as a balance: large enough to allow meaningful adaptation, but not so large as to destabilize training. Still, we found that the rate and extent of perplexity change varied significantly across domains, reinforcing the difficulty of defining a one-size-fits-all schedule.

