# OpenReview forum: "Investigating Continual Pretraining in Large Language Models: Insights and Implications"
_TMLR — Accepted by TMLR_

### Review · Reviewer_dnFi · 2025-02-27

**Summary Of Contributions:**

This paper presents a comprehensive empirical study of continual domain-adaptive pretraining for large language models (LLMs). The authors introduce a new benchmark based on the M2D2 dataset, enabling a more realistic evaluation of knowledge retention and transfer across diverse domains than existing continual learning benchmarks. Through extensive experiments across various model sizes and architectures (GPT-2 family, Llama2-7B, RoBERTa), they uncover several key insights about continual pretraining, including its effectiveness compared to domain adaptation, the impact of model size, the influence of training domain order, and the resulting downstream task performance. The paper provides a valuable characterization of the challenges and opportunities associated with continual pretraining in LLMs, contributing significantly to the understanding of this important area.

**Audience:**

Yes

**Claims And Evidence:**

Yes

**Requested Changes:**

I think the paper would benefit from improving the training details especially the meaning of what each of the compared "baselines" mean in implementation along with the final learning rates.

The pretrained model's learning rate and schedules affect the continual pre-training results a lot and would be helpful to keep in perspective when explaining things to the readers.

**Strengths And Weaknesses:**

Strengths:

Comprehensive Empirical Study: The paper's primary strength lies in its extensive empirical evaluation. The authors meticulously conduct experiments across a wide range of models, tasks, and training configurations, providing a wealth of data to support their claims.

Novel Benchmark: The introduction of a new benchmark based on the M2D2 dataset is a valuable contribution. This benchmark addresses limitations of existing continual learning benchmarks by operating at a more realistic scale for LLMs and encompassing a diverse set of domains.

Key Insights: The paper identifies and articulates several valuable insights about continual pretraining. These insights are well-supported by the experimental results and provide practical guidance for researchers and practitioners in the field.

Clear Presentation: The paper is generally well-written and easy to follow. The experimental setup, evaluation metrics, and results are clearly presented, making the paper accessible to a broad audience.


Weaknesses:

Limited Background on Domain-Adaptive Pretraining: While the paper focuses on continual domain-adaptive pretraining, it doesn't explicitly define or describe the specific techniques used for domain-adaptive pretraining itself. Some readers may benefit from a brief overview of common domain adaptation methods employed during pretraining.

Lack of Discussion on Learning Rate Selection: The learning rate is a critical hyperparameter in LLM training, especially in continual learning scenarios where the model is adapting to new domains. The paper mentions automatic learning rate selection via DeepSpeed but doesn't provide details about the chosen learning rate range -- this is crucial for having the results put on a fair comparison perspective.. A brief discussion of learning rate choices and their potential impact on learning and forgetting would be valuable.

-----------------
Overall I agree with the takeaways of the paper and think it is of value to the community to know these takeaways.

---

> ### Author Response · Authors · 2025-03-11
> **start of discussion**
>
> Thanks for the insightful comments.
>
> 1. In this work, we deliberately adopted the simplest form of domain-adaptive pretraining (DAPT), which is continuing to pretrain a base model on data from a single domain, without including any other domains. Our goal was to establish a clear and interpretable baseline for continual pretraining performance. While this setup does not involve more complex techniques such as adapters, domain tags, or multi-domain interpolation, we believe it is representative of standard DAPT approaches used in prior work (e.g., Gururangan et al., 2020). We will add a brief discussion in the revised manuscript to clarify this choice and distinguish our approach from alternative DAPT strategies.
> 2. This is an excellent point, and we appreciate the opportunity to expand on it. In our experiments, we used a linear warmup learning rate schedule, increasing from 0.0 to 5e-5 for GPT2 models and 0.0 to 1e-5 for Llama2-7B, over 250 warmup steps, after which the learning rate remains constant. These values were chosen based on common practice in pretraining and our preliminary tuning.
> We acknowledge that learning rate selection plays a particularly critical role in continual pretraining, where subtle changes in optimization dynamics can significantly affect both knowledge acquisition and forgetting. In our experiments:
>   - For GPT2 models, we consistently observed that both training and test perplexities decreased, especially in larger domains (>50MB). Higher learning rates sometimes helped smaller domains converge faster, though not always consistently. However, we found that training loss behavior on the current domain did not reliably predict test perplexity behavior on other domains. In other words, sometimes rapid convergence on one domain led to performance degradation on others, and vice versa. These observations suggest that there is no universal or robust learning rate recipe for continual pretraining across heterogeneous domains.
>   - For Llama2-7B, learning rate sensitivity was even more pronounced. Smaller learning rates often resulted in almost no model updates, while larger learning rates led to both larger improvements and larger degradations, depending on the domain. We settled on 1e-5 as a balance: large enough to allow meaningful adaptation, but not so large as to destabilize training. Still, we found that the rate and extent of perplexity change varied significantly across domains, reinforcing the difficulty of defining a one-size-fits-all schedule.
>
> We will revise the paper to include a dedicated discussion on these findings, including the learning rate values used, the rationale behind them, and the observed implications for continual pretraining performance.

---

> > ### Comment · Reviewer_dnFi · 2025-03-11
> >
> > Thanks for the clarifications. Would love to see these changes in the next manuscript.

---

### Review · Reviewer_XJRM · 2025-03-06

**Summary Of Contributions:**

This paper introduces a new benchmark to assess the ability of LLMs to continually learn. Concretely, this paper focuses on the **continual domain-adaptive pretraining case**, where a base LLM is continually pretrained further on multiple new domains in a sequential fashion. To that end, the paper explores continual domain-adaptive pretraining on three models: The GPT-2 family of models (of various sizes), Llama2 (with 7 billion parameters), and also RoBERTa, with an emphasis on measuring both forward (i.e. generalization to new domains) & backward transfer (i.e. model forgetting).

Through experiments on various model sizes and 236 domains, the paper finds the following insights:
- Continual pretraining helps different-sized models to different extents, whereby smaller models (<1.5B) acquire the most additional knowledge, albeit also suffer the most from forgetting;
- Overall, larger models still achieve the best perplexity, even though the improvements (and forgetting) from continual pretraining tend to be less substantial than the improvements for smaller models;
- Continual learning performance, as measured by language modeling perplexity, is indicative of downstream task performance on related domains;
- The **curriculum** under which continual pretraining is done affects both transfer & final performance. To that end, conducting continual pretraining where new domains are added in a *random* fashion enables positive transfer and reduces forgetting, whereas presenting the new domains based on their similarity to previous domains improves performance more on related domains, albeit at the expense of less domain transfer to unrelated domains, stronger forgetting, and sometimes worse final performance.

**Audience:**

Yes

**Broader Impact Concerns:**

No broader impact concerns come to mind.

**Claims And Evidence:**

Yes

**Requested Changes:**

1. **Critical**: Address the first weakness, which is that some of the findings are either well-known, obvious, or not particularly novel.

2. **Critical**: Add a baseline where we don't have to do continual pretraining in a sequential fashion, and can simply mix in all the new domain data and then train on it in an i.i.d. fashion (second weakness).

3. **Critical**: Fix the overloaded notation (third weakness).

4. **Critical**: Redo the experiment on Section 4.6 with one of the GPT2 models (rather than Llama2-7B which does much worse due to continual pretraining).

5. **Recommended**: Dive deeper on why the Llama2-7B continual pretraining performance is bad, for example by using other hyper-parameters, etc.

6. **Recommended**: Bring some of the Appendix figures to the main paper to make it more self-contained.

7. **Recommended**: Address the questions & the big picture research suggestions.

**Strengths And Weaknesses:**

# Strengths

1. Continual learning in LM pretraining is an important research area, and how we can inject additional knowledge into those models, while at the same time minimizing catastrophic forgetting, remains an important open research question. This paper makes progress towards answering some open questions in this space through more comprehensive continual LM pretraining benchmarking than what has been done in prior work.

2. Some of the insights from the paper, particularly relating to (i) the best continual pretraining curriculum to maximize forward transfer (e.g. that presenting the domains in a random order is very competitive in terms of forward transfer and final performance, and should therefore be the default option); (ii) the minimum amount of data to successfully fine-tune different-sized models (and how some domains are too small to fine-tune the 7B-parameter Llama 2 model); and the fact that (iii) smaller LMs exhibit the most continual learning (i.e. the acquisition of new knowledge, as indicated by the largest performance improvement relative to the zero-shot baseline), while at the same time also exhibiting the most forgetting), would be useful for the broader community.

3. The paper is overall well-written and very easy to follow.

# Weaknesses

1. Some of the findings are either well-known or not particularly novel, which would not be very useful for the community. For example, the fact that larger models perform better in terms of final performance (compared to smaller models, Section 4.2) is already well-known for a few years (Kaplan et al., 2021) and demonstrated many times over at this point. Furthermore, the fact that perplexity correlates very well with downstream task on related domains (Section 4.5) is also not particularly surprising, and is already well-established in the field. Lastly, the fact that positive forward transfer in similar training order is possible to only semantically related domains (Section 4.4) is also very intuitive and not surprising (basically under this semantically-related continual pretraining curriculum, the model can specialize hard on multiple new domains that are semantically very related to each other, which means that they also exhibit more forgetting & less transfer to older / unrelated domains).

2. The paper is missing what in my opinion is an important baseline: The case with no continual learning at all, where the 236 new domain data are simply mixed together, and concatenated with the pretraining data. This setting can indicate the **upper bound on model performance**, where the model does not have to learn new domains in a sequential fashion, and can also potentially do multiple epochs over the new data. Basically: How much do we lose in the continual pretraining setup, as opposed to the case where we can just mix in all the new data together without having to do continual learning in a sequential fashion?

3. I have concerns with the notation in page 4, Section 3. My main concern is that the FG / forgetting metric, which is defined as the median of $f_2, \cdots, f_N$, crucially **overloads** the notation for $f$, which is already used for $f_n$ above it, which "denotes the zero-shot and domain adaptive pretraining perplexities computed on the $n$-th domain". This means that the symbol $f$ now has two overloaded meanings, which includes the $f_c$ definition used for computing FG / forgetting.

4. Section 4.6, which computes the prediction rank-based analysis for knowledge accumulation, is done using the Llama2-7B model. This choice sounds questionable to me, because the Llama2-7B model does not seem to learn much at all from the continual pretraining stage, which worsens its performance by a big margin. I would suggest redoing these experiments with one of the GPT2 model sizes instead.

5. The continual pretraining results with Llama2-7B seem to be quite bad for the vast majority of domains. Could this be a learning rate / batch size issue? In other words, how much hyper-parameter tuning is done to find a good set of continual pretraining parameters for Llama2 7B? This is particularly surprising given the fact that training perplexity always improved (Page 5, Section 4.1). Are the authors saying that this is an overfitting issue?

6. There seems to be a pretty heavy reliance on the Appendix, where some Figures that are pretty central to e.g. Section 4.5 are in the Appendix (e.g. Figures 11 & 12). It'd be nice to have all the important figures in the main paper, which would make the paper more self-contained.

# Questions & Suggestions & Comments
1. For the DAPT baseline (page 4, Section 3), does it start from a model that has been pretrained? Or does it always pretrain on each domain from scratch?

2. On page 6, there is a mention to "... similar-order training (Figure 5, **right** ) ...". This could be a typo, because the similar order training is in Figure 5, **left**.

3. Pedantic comment: The model sizes explored in this work are pretty small (up to 7B). I'm not holding this against the paper because running larger LMs are very expensive, but I am not sure if they qualify as LLMs, which stand for **large** language models.

4. On page 6, it is mentioned that "increasing model size could be a way to alleviate forgetting...". But at the same time, larger models also don't acquire as much new knowledge in the continual pretraining stage as smaller models (though they still perform better in terms of final accuracy). Given that learning & forgetting are intertwined, I'm not sure how much the recommendation to use larger models is a good conclusion.

5. Typo: The first sentence of Section 4.3 is lacking a verb (the subject is "Figures 3-5 ..." but there is no corresponding verb, maybe something like "showcase / demonstrate" or something like that?)

6. For figure 9, mention that "higher is better" or something like that to contrast with the perplexity evaluations, where lower is better.

7. For the batch size experiments (page 10), are the learning rates adjusted accordingly (i.e. increased learning rate when the batch size is increased and we do fewer updates?).

8. Related work: Mind the Gap: Assessing Temporal Generalization in Neural Language Models (Lazaridou et al., 2021), which also proposed a benchmark for continual learning of LMs.

# Big Picture Suggestions
Given that some of the findings are already known / less novel, I would suggest exploring two additional research directions to increase the impact of the paper:
1. Parameter averaging has been shown as a cheap way to do model combination, which would work here because we start from the very same base model. What happens if you average the parameters after training on each domain? For example, if there are $M$ domains and after doing continual pretraining on each domain, we have $M$ saved checkpoints, we can simply average these checkpoints in the parameter space. My guess is that this would mitigate the forgetting issue.

2. One thing that would be very useful is training a model to predict: How much improvement & forward / backward transfer will there be from doing continual pretraining on a particular domain? This can tell us which data will bring the most value if we train on them, while at the same time allowing us to focus our efforts on identifying data that will be most useful for improving performance on a given dataset / domain.

---

> ### Author Response · Authors · 2025-03-11
> **start of discussion**
>
> We thank the reviewer for their elaborate and deeply insightful comments. Below are our responses:
>
> ### Weaknesses
> (Please note that our responses also cover “requested changes”)
>
> 1. We appreciate the reviewer’s comments here. Regarding scaling laws, while it is indeed well-known that larger models perform better in general (e.g., Kaplan et al., 2020), to the best of our knowledge, our work is the first to systematically assess this behavior in the context of continual domain-adaptive pretraining across multiple model sizes and architectures in a high-resolution benchmark setting. If the reviewer is aware of prior work demonstrating this specifically in continual pretraining scenarios, we would be happy to cite it.
> As for the perplexity–downstream performance correlation, we agree that this is intuitive. However, LLMs are often evaluated exclusively via downstream tasks, and we felt it was important to explicitly validate this connection in the continual pretraining context. Interestingly, our results also show that Llama2-7B achieves reasonable perplexity (≈11) but performs at chance level on downstream tasks, highlighting a potential mismatch between standard perplexity-based metrics and practical task performance. This, we believe, is an important observation that warrants inclusion.
> We also agree that positive forward transfer in similar-order training is somewhat expected. Nonetheless, we believe our experiments offer value by quantifying how much transfer occurs, to which domains, and to what extent it degrades over time, which helps contextualize the dynamics of domain similarity in a measurable way.
> 2. We completely agree this is a valuable baseline. We have just initiated such an experiment and aim to include it in the updated version. This will serve as an upper bound benchmark where all new domains are mixed and trained in an i.i.d. fashion, thus offering a direct comparison point to our continual pretraining setup.
> 3. Thank you for pointing this out. We agree the notation in Section 3 needs clarification. We will revise this section.
> 4. Thank you for highlighting this concern. First, we would like to note that we already conducted the same analysis with GPT2-M, and the results are provided in Figure 14. We apologize that this figure might have been overlooked — we will move it into the main paper to highlight it more prominently.
> The motivation for Section 4.6 stems from the limitations of downstream task evaluation, especially for Llama2-7B, which, despite reasonable perplexity, performs poorly on downstream tasks (Section 4.5) when prompted zero/one/few shot. The rank-based method provides an alternative metric for assessing domain knowledge retention, independent of prompting ability. We observe improving/degrading performance for GPT2-M/Llama2-7B, respectively, which is aligned with our perplexity findings. More importantly, ranks achieved by Llama2-7B show a much more interpretable pattern than the downstream task performance: we see the continual degradation while occasional improvements due to training domains.
> 5. We were indeed surprised by Llama2-7B’s poor continual pretraining performance. We used default pretraining parameters from the Llama 2 paper, with the exception of using AdamW, 250 warmup steps from learning rate 0.0 to 5e-5. While training perplexity always decreased, downstream performance degraded, especially in in-context learning.
> We believe this may not be traditional overfitting but rather an optimization brittleness, possibly arising from small domain sizes. As shown in Figure 17, performance improves quite a bit (perplexity going from ~35 to 7) with larger domains (e.g., 100MB vs. 50MB). We also observed that perplexity on frequent tokens (e.g., "the", "\n") degrades heavily, which impacts overall evaluation. This phenomenon may be tied to catastrophic interference during gradient updates, which is increasingly noted in LLMs during continual learning and which often requires plenty of gradient updates to recover the base performance. We are continuing to investigate this behavior.
> 6. Agreed. We will move several key figures (e.g., Figures 11, 12, and 14) into the main paper to make the narrative more self-contained and easier to follow.

---

> ### Author Response · Authors · 2025-03-11
> **continuation of discussion**
>
> ### Questions
> 1. Yes, DAPT always starts from the original base model, not from previous domains.
> 2. Thank you, we will correct this.
> 3. We appreciate the comment. While we agree that 7B may not be "large" by today's standards, in the continual learning research context, it remains on the higher end of model scales.
> 4. Excellent point. As you note, larger models may forget less because they also acquire less new knowledge in this setting. We will make this point more explicit in our discussion.
> 5. Thanks, will fix it.
> 6. Noted, will clarify.
> 7. No, we used the same learning rate for all batch sizes. We’ll clarify this in the revision.
> 8. Thanks, we will add it to our references.
>
> ### Big picture suggestions
> 1. This is an excellent suggestion. We have started testing parameter averaging of continual checkpoints to investigate whether it can mitigate forgetting. Depending on experiment completion time, we hope to include this in the final revision or note it as a promising future direction.
> 2. Another nice idea! We have done preliminary experiments along these lines but struggled with domain representation and transfer quantification. While bandit-style heuristics show potential (e.g., a checkpoint that was trained on cs.AI transfers better to cs.CV than math.CA), we found it challenging to measure “expected transfer” in a generalizable way. This is a direction we’re excited to explore further and will mention as future work.
>
> ### Requested changes:
> We hope that our above answers also address requested changes 1,2,3,4,6. We would be happy to hear reviewer’s thoughts on our idea “why the Llama2-7B continual pretraining performance is bad” (requested change 5) and discuss this interesting point more. Finally, for requested change 6, we are happy to execute the weight averaging experiment if we have sufficient resources (we prioritize the “mixed data points” experiment as well as another similar-order experiment raised by other reviewers)

---

> > ### Comment · Reviewer_XJRM · 2025-03-24
> > **Thank You For The Response**
> >
> > I would like to thank the authors for their thorough authors' response, which has addressed most of my concerns. I have also read the other reviews (some of which raised similar concerns), and also the authors' response to the other reviews.
> >
> > In my view, the authors' response and the revised version strengthens the work. Overall, I think this paper will be useful for the broader community.

---

### Review · Reviewer_F7Li · 2025-03-07

**Summary Of Contributions:**

This paper presents an experimental study of continually pre-training a set of tasks. Experiments are run on benchmark datasets involving a large number of tasks and several domains. Several pre-training architectures are tested. This experimental paper provides conclusions around the behavior of continual pre-training under different architectures, for different tasks, for different model sizes. It investigates the roles of forgetting, backward transfer and forward transfer.

**Audience:**

Yes

**Broader Impact Concerns:**

No broader impact concerns.

**Claims And Evidence:**

Yes

**Requested Changes:**

Please refer to weaknesses. I would like to see multiple task orders included in the experiments, and also (subsampled) datasets beyond 5GB. Also, please fix the explanation of metrics and how you use these to refer to results in the experiments.

**Strengths And Weaknesses:**

Strengths:
- The paper definitely explores an interesting problem from an experimental perspective.
- The conclusions, such as for example those associated with model size or with different architectures, are potentially very useful for researchers in the field.
- The paper is well-written and well structured. The experiments are wide enough for the type of conclusions made thorough the paper.

Weaknesses:
- As acknowledged in the conclusion, experiments were run for a single task order. This definitely impacts the results, as different tasks orders will affect forgetting, backward and forward transfer differently. Without this, it is hard to consider the results generalizable.
- Related to the previous point, the "dissimilar" task order was ignored.
- The writing of the metrics in section 3 is a bit confusing, as there are many concepts (i.e. metrics) explained at once. I believe this could be better structured. Also, later on in the experiments, connecting the things that are actually being measured to the definition of these metrics in section 3 is hard, since no acronyms are used in figures for example.
- Finally, it is also mentioned that experiments were limited to domains below 5GB. Why not simply subsampling these large domains to cover more variety?

---

> ### Author Response · Authors · 2025-03-11
> **start of discussion**
>
> We thank the reviewer for their insightful comments. Below are our responses:
>
> 1. Thank you for the feedback. We agree that task order can influence continual learning dynamics such as forgetting and transfer. To assess the robustness of our findings, we reran our random-order experiments with two additional random permutations (Section 4.7: “Alternative random orders yield similar findings”). The results were consistent: median CPTs of 16.4 and 16.78, compared to 16.82 in the main run; forgetting scores of 1.72 and 1.85, compared to 1.78 in the main run.
> For similar-order training, we also conducted an alternative ordering by reversing the Wiki and S2ORC portions. As shown in Figure 21, the key trends remain consistent, confirming the robustness of our main insights. That said, we acknowledge that evaluating only a single similarity-based order is a limitation, and we are currently running additional similar-order variants to strengthen generalizability.
> 2. Our design intentionally contrasts similar-order and random-order regimes, motivated by real-world analogies, e.g., curriculum learning vs non-i.i.d. internet-scale data streams. However, we are unsure what would constitute a well-motivated and principled "dissimilar" task order. If the reviewer has a specific definition or intended use case in mind, we would be happy to consider it if we are provided with more time to conduct the experiment because of the high computational cost involved in training.
> 3. Thank you for pointing this out. We agree the presentation of metrics in Section 3 could be clearer. We will revise this section to improve structure and clarity by grouping related metrics and simplifying notation.
> In addition, we will ensure that acronyms (e.g., CPT, LC, FG) are clearly annotated in all relevant figures and captions, making it easier to connect definitions to empirical results throughout the paper.
> 4. This is a fair point. We excluded domains larger than 5GB primarily due to computational constraints, not due to any methodological limitation. We aimed to include as broad a variety as feasible given training budget and evaluation overhead (our benchmark includes 159 domains, each checkpoint evaluated on all prior domains).
> That said, we agree that incorporating additional domains (even via subsampling) could further enrich the analysis. We are now evaluating whether we can include a few additional subsampled domains within the rebuttal timeline. Although due to time and resource limits, we may need to defer this to the camera-ready version.

---

### Review · Reviewer_QXyD · 2025-03-08

**Summary Of Contributions:**

This paper investigates the impact of continual pretraining in large language models (LLMs), distinguishing itself from existing works that primarily focus on continual fine-tuning. The study introduces a new benchmark to measure the adaptability of LLMs to dynamic pretraining data landscapes. By systematically analyzing continual pretraining dynamics, the paper examines knowledge retention, forward and backward transfer, and forgetting across various models. This paper leverages multiple evaluation metrics to provide insights into the efficacy of CL in LLMs. Moreover, it explores the impact of model architecture (decoder-only vs. encoder-decoder), domain ordering strategies, and model size on continual pretraining outcomes.

**Audience:**

Yes

**Broader Impact Concerns:**

No broader impact concerns.

**Claims And Evidence:**

Yes

**Requested Changes:**

1. Could authors please add tables to describe dataset arrangements in similar-order and random-orders? For example, in similar-order 1: 1. Computer Science -> 2. Physics -> 3.... Providing such details would improve clarity regarding task sequencing.
2. Could authors please add more explanations on how to determine the ordering of the training domains based on their similarity? How is the first task selected? Additionally, how many different random-orders are explored in this paper?
3. About Section 4.6, could authors explain why this analysis is relevant to continual learning insights? Additionally, how do the results of rank-based analysis support the findings on knowledge retention and transfer? This subsection lacks a clear explanation of its relevance to the paper, making it difficult to understand its significance within the broader context of the study.

**Strengths And Weaknesses:**

Strengths:
1. This paper provided a detailed analysis of continual pretraining across a broad range of domains using the M2D2 dataset, offering a large-scale benchmark for LLM adaptability.
2. Findings on different decoder-only models (GPT2-S, GPT2-M, GPT2-L, GPT2-XL, Llama2-7B) and different encoder-decoder models (RoBERTa-base and RoBBERTa-large) provide useful insights into improving continual learning strategies.
3. The use of different and diverse evaluation metrics, including perplexity, forgetting, forward and backward transfer, and downstream performance, provides a comprehensive assessment of CL in LLMs.

Weaknesses:
1. Some key findings in this paper are not first proposed, such as “larger models always achieve better perplexity than smaller ones when continually pretrained on the same corpus”. Additionally, some findings may be too specific for individual model architectures, like Llama2-7B, and lack broader generalizability.
2. The principle of designing the similar order is not clear to me. Moreover, this paper does not sufficiently clarify how tasks are positioned in different random-order settings. Besides, the authors mentioned in the limitations section that “in similar-order training, domains were ordered only once, meaning our findings may not generalize to other orderings”, but exploring different domain orders is crucial in continual learning, if similar-order only uses only a single specific order, the findings in similar-order may limit the generalizability.
3. In the experiments, the authors primarily focus on comparing different models with the proposed random-order and similar-order, but I did not find the comparison with some baselines of other domain-adaptive pretraining approaches. Including some baselines would help contextualize the efficacy of the proposed benchmark more comprehensively.

---

> ### Author Response · Authors · 2025-03-11
> **start of discussion**
>
> We thank the reviewer for their insightful comments. Below are our responses:
>
> ### Weaknesses
> 1. To the best of our knowledge, the observation that "larger models consistently achieve better perplexity than smaller ones when continually pretrained on the same corpus" has not been formally established in the context of continual domain-adaptive pretraining. While this is an intuitive result aligned with existing scaling law literature (e.g., Kaplan et al., 2020; Bahri et al., 2021), we are not aware of prior work that explicitly demonstrates this trend under continual pretraining conditions. If the reviewer has references on this, we would be happy to incorporate them.
> Importantly, our paper contributes more than this particular observation. For example, we show that (i) larger models forget less during continual pretraining, (ii) the smaller models improve more, and (iii) how the model size impacts backward transfer (as a function of transfer distance). We believe our results offer nuanced insights beyond simply confirming scaling laws, especially in the continual learning setting.
> 2. Thanks for the request for clarification. In random-order training, we shuffle the 159 L2 domains and conduct continual pretraining based on this permutation. To test robustness, we re-ran experiments with two additional random orderings (Section 4.7: "Alternative random orders yield similar findings") and found the results to be consistent (median CPT: 16.4 and 16.78 vs. 16.82 in the main run; also forgetting: 1.72 and 1.85 vs. 1.78 in the main run).
> For similar-order training, the domain ordering is determined via cosine similarity to OpenWebText embeddings, starting with the most similar (Culture) and proceeding by sampling next domains based on similarity (Section 2; Fig 1). Additionally, we conducted a swap experiment (reversing the Wiki and S2ORC portions in the similar-order sequence) to evaluate robustness, and verified that the key findings remain consistent (see Figure 21). Nonetheless, we agree that relying on a single similarity-based sequence is a limitation and we are currently running additional similar-order variants to verify generalizability.
> 3. This is a valuable point. In our setup, domain-adaptive pretraining (DAPT) is a core baseline, and we compare continual pre-training perplexity (CPT) and final checkpoint performance against it throughout (see Section 3 and Figures 3, 5, 6). To our knowledge, most DAPT approaches involve sequential adaptation on individual domains, which is effectively what we benchmark against in our experiments. That said, if the reviewer is referring to more sophisticated DAPT approaches such as mixture-of-experts or adapter-based pretraining, we would leave them as interesting comparisons for future work that build actual continual learning algorithms.
> Furthermore, we are conducting new experiments where all training data points are mixed and used for full adaptation, providing an upper bound on continual pretraining performance. We will include these results in the revised manuscript to better contextualize our findings.

---

> ### Author Response · Authors · 2025-03-11
> **requested changes**
>
> 1. Thank you for the suggestion. We will include a detailed table in the appendix listing the complete L2 domain sequences used in both similar-order and random-order training runs. For the similar-order configuration, the ordering directly corresponds to the rows in Figure 1 and is derived from cosine similarity to OpenWebText embeddings. This should help clarify the curriculum structure used in our study.
> 2. As described in Section 2 (“Tasks” paragraph), similar-order training begins with the domain most similar to OpenWebText (Culture), followed by probabilistic sampling of the next domain from those most similar to the current one. Cosine similarity between domain embeddings, which were computed using Sentence-BERT, is used as a proxy for semantic proximity (see Figures 1 and 2).
> Regarding the number of random orders, we use one primary random order to ensure consistency across experiments. To test generality, we include two additional random shufflings (Section 4.7), which yielded consistent results, confirming that findings are not sensitive to a particular permutation.
> 3. Thank you for raising this important question. LLMs are almost always benchmarked on some downstream task rather than reporting raw perplexity numbers, which depend on vocabulary size, tokenization, etc. We included a benchmark study already in Section 4.5. The motivation behind Section 4.6 stems from limitations we observed in the said downstream evaluation, especially for continually pretrained Llama2-7B checkpoints, which struggle with zero-/one-/few-shot prompting.
> Our proposed rank-based method offers an alternative way of measuring whether domain-specific concepts remain predictable in context, even when downstream task metrics fail. The method evaluates token prediction ranks for domain-specific keywords in sentence contexts, enabling us to trace knowledge transfer across domains (Figure 10).
> Notably, we find that this analysis provides insights into knowledge accumulation and forgetting in LLMs. We observe improving/degrading performance for GPT2-M/Llama2-7B, which is aligned with our perplexity findings. More importantly, ranks achieved by Llama2-7B show a much more interpretable pattern than the downstream task performance, which was chance-level for almost all checkpoints. We will revise the section to make this motivation and its relevance to continual learning clearer.

---

### Author Response · Authors · 2025-03-11
**Thanks to reviewers and general message**

We would like to thank all reviewers for their insightful comments and acknowledging (i) the comprehensive, timely, and large-scale nature of our study, (ii) the rich analysis using multiple evaluation metrics, (iii) practical insights on model behavior, (iv) clarity and readability of the paper, and (v) our useful comparisons across model architectures and sizes.
In this general response, we would like to summarize the main changes until the rebuttal time. We are hoping that we will have enough time to conduct the experiments mentioned below:
- Add a detailed table listing the domain sequences
- Clarify the procedure used for constructing similar-order training sequences
- Add a new experiment with an alternative similar-order variant
- Introduce a new baseline experiment using fully mixed domain data (non-continual i.i.d. training)
- Clarify domain-adaptive pretraining (DAPT) baseline implementation details
- Revise and improve the explanation of evaluation metrics in Section 3
- Fix overloaded notation issues
- Move key figures (e.g., Figures 11, 12, and 14) from the appendix into the main paper
- Highlight results of the rank-based knowledge accumulation analysis using GPT2-M
- Provide a clearer justification and explanation for the rank-based analysis in Section 4.6
- Discuss learning rate choices in more detail
- Add training details (more explanation on the implementation of the baselines)
- [if time permits] Consider adding additional domains via subsampling from domains >5GB
- [if time permits] Explore a parameter averaging experiment

---

> ### Author Response · Authors · 2025-03-21
> **Final changes**
>
> Thanks again all reviewers for their time creating insightful feedback. We modified the text and ran additional experiments to address the reviewers' points. Whenever possible, we updated the main text. Otherwise, we added supplementary explanations to the appendix. In light of the reviews and potential new comments, we would like to move some figures and text from the appendix to the main body, which we hope to do in the camera-ready version. Below is a summary of our changes:
>
> Added Appendix A3 to address:
> - Clarify the procedure used for constructing similar-order training sequences (see "How did we order domains?" paragraph)
> - Add a new experiment with an alternative similar-order variant (see "Additional similar order experiments" paragraph)
> - Add a detailed table listing the domain sequences (see the remaining paragraphs)
>
> Added Figure 11 to address:
> - Introduce a new baseline experiment using fully mixed domain data (non-continual i.i.d. training)
>
> Updated sections 2 and 3 by moving all the notation to a new "Metric expressed explicitly" paragraph at the end of section 3 and rephrasing certain passages to address:
> - Clarify domain-adaptive pretraining (DAPT) baseline implementation details
> - Revise and improve the explanation of evaluation metrics in Section 3
> - Fix overloaded notation issues
> - Add training details (more explanation on the implementation of the baselines)
>
> Added a first paragraph to Section 4.6 to address:
> - Provide a clearer justification and explanation for the rank-based analysis in Section 4.6
>
> We updated the last paragraph of Section 4.6 to address:
> - Highlight results of the rank-based knowledge accumulation analysis using GPT2-M
>
> We added subsection A.4:
> - Discuss learning rate choices in more detail
>
> We realized that this would lead our submission to go beyond 12 pages. We will check with the action editor before making the following change:
> - Move key figures (e.g., Figures 11, 12, and 14) from the appendix into the main paper
>
> We unfortunately did not have resources to perform the following experiments:
> - Consider adding additional domains via subsampling from domains >5GB
> - Explore a parameter averaging experiment

---

> > ### Comment · Reviewer_dnFi · 2025-03-21
> >
> > Thanks for the changes and additional details. I am supportive of the paper acceptance.

---

> > ### Comment · Reviewer_QXyD · 2025-03-21
> >
> > I appreciate the authors' efforts in addressing my concerns. The revised version of the paper resolves most of the issues I raised. It would be more appropriate to present the list of domain sequences in Appendix A3 as a LaTeX table rather than as a text paragraph. Overall, I am pleased to recommend acceptance.

---

> ### Comment · Reviewer_F7Li · 2025-03-29
> **Supporting acceptance**
>
> Thanks to the authors for the insightful responses and changes. I am also supportive of the paper acceptance.

---

### Decision · Action_Editor_D3af · 2025-04-18

**Recommendation:** Accept with minor revision

**Comment:**

The paper presents a thorough empirical study on continual domain-adaptive pretraining for language models, supported by a newly introduced benchmark and a wide range of experiments. Reviewers highlighted the depth and breadth of analysis, the value of the insights, and the clarity of presentation. The authors responded constructively to all critical comments and improved the paper accordingly. While some findings confirm expected behavior (e.g. larger models forget less), the contribution remains valuable for establishing best practices and a baseline in a relatively underexplored area.
Recommend the authors implement the minor revisions suggested during the review process.

**Audience:**

The paper is relevant to TMLR’s audience, especially those interested in continual learning, language modeling, and large-scale empirical studies. The new benchmark and practical insights into continual pretraining are likely to benefit both academic and applied research communities.

**Claims And Evidence:**

The claims are well supported through extensive empirical evaluation across multiple model families, pretraining settings, and evaluation metrics. The authors address reviewer concerns in detail and provide new results, including robustness checks, additional baselines, and clarified experimental procedures.

---

> ### Author Response · Authors · 2025-04-28
>
> Dear Reviewers and Action Editors,
>
> Thank you for your thoughtful feedback and support throughout the review process.
>
> We have implemented the requested changes, including adding author information and acknowledgements, and have carefully revised the paper to address all suggestions. The final camera-ready version has been expanded to 12 pages, incorporating the minor revisions and improvements discussed. We have submitted the final version accordingly.
>
> We sincerely appreciate your valuable feedback, which helped us strengthen the paper.
>
> Best regards,
>
> TMLR Paper4174 Authors